# The metalloproteinase Papp-aa controls epithelial cell quiescence-proliferation transition

Chengdong Liu[1], Shuang Li[1,2], Pernille Rimmer Noer[3], Kasper Kjaer-Sorensen[3], Anna Karina Juhl[3], Allison Goldstein[1], Caihuan Ke[2], Claus Oxvig[3]*, Cunming Duan[1]*

[1]Department of Molecular, Cellular, and Developmental Biology, University of Michigan, Ann Arbor, United States; [2]College of Ocean and Earth Sciences, Xiamen University, Xiamen, China; [3]Department of Molecular Biology and Genetics, Aarhus University, Aarhus, Denmark

**Abstract** Human patients carrying *PAPP-A2* inactivating mutations have low bone mineral density. The underlying mechanisms for this reduced calcification are poorly understood. Using a zebrafish model, we report that Papp-aa regulates bone calcification by promoting $Ca^{2+}$-transporting epithelial cell (ionocyte) quiescence-proliferation transition. Ionocytes, which are normally quiescent, re-enter the cell cycle under low $[Ca^{2+}]$ stress. Genetic deletion of Papp-aa, but not the closely related Papp-ab, abolished ionocyte proliferation and reduced calcified bone mass. Loss of Papp-aa expression or activity resulted in diminished IGF1 receptor-Akt-Tor signaling in ionocytes. Under low $Ca^{2+}$ stress, Papp-aa cleaved Igfbp5a. Under normal conditions, however, Papp-aa proteinase activity was suppressed and IGFs were sequestered in the IGF/Igfbp complex. Pharmacological disruption of the IGF/Igfbp complex or adding free IGF1 activated IGF signaling and promoted ionocyte proliferation. These findings suggest that Papp-aa-mediated local Igfbp5a cleavage functions as a $[Ca^{2+}]$-regulated molecular switch linking IGF signaling to bone calcification by stimulating epithelial cell quiescence-proliferation transition under low $Ca^{2+}$ stress.

*For correspondence:
co@mb.au.dk (CO);
cduan@umich.edu (CD)

Competing interests: The authors declare that no competing interests exist.

## Introduction

Pregnancy-associated plasma protein-a (PAPP-A) and PAPP-A2 belong to the conserved pappalysin proteinase family (*Boldt et al., 2001*; *Oxvig, 2015*; *Conover and Oxvig, 2018*). PAPP-A was first discovered in the plasma of pregnant women in the 1970s. Decades of studies suggest that PAPP-A and PAPP-A2 are zinc metalloproteinases cleaving insulin-like growth factor binding proteins (IGFBPs) (*Laursen et al., 2007*; *Oxvig, 2015*; *Conover and Oxvig, 2018*). IGFBPs are a family of six secreted proteins that bind IGF ligands with high affinity and regulate IGF availability to the IGF1 receptor (*Baxter, 2014*; *Allard and Duan, 2018*; *Clemmons, 2018*). In vitro studies suggested that PAPP-A is tethered to the cell-surface and primarily cleaves IGFBP4 and IGFBP5 among the six IGFBPs. PAPP-A2, on the other hand, is secreted and cleaves IGFBP3 and IGFBP5 (*Oxvig, 2015*). Papp-a knockout mice showed a 40% reduction in body size (*Conover et al., 2004*), a phenotype similar to those of the IGF1 and IGF2 mutant mice (*Baker et al., 1993*; *Liu et al., 1993*). Papp-a2 knockout mice had a modest decrease in body size (*Christians et al., 2013*; *Christians et al., 2019*). These findings led to the proposal that PAPP-A and PAPP-A2 promote somatic growth by increasing bioavailable IGFs (*Oxvig, 2015*; *Fujimoto et al., 2017*; *Conover and Oxvig, 2018*). This notion is further supported by recent clinical studies showing that human patients carrying inactivating mutations in the PAPP-A2 gene displayed progressive growth failure (*Dauber et al., 2016*; *Fujimoto et al., 2017*). In addition to reduced body height, these patients also had lower bone

mineral density (*Dauber et al., 2016*; *Fujimoto et al., 2017*). Likewise, the Papp-a mutant mice had delayed appearance of ossification centers and reduced calcified bone mass (*Conover et al., 2004*). The Papp-a2 mutant mice had sex-and age-specific defects in bone structure and mineral density (*Christians et al., 2019*). Despite observations suggest a role of these metalloproteinases in bone mineralization, there is still a lack of causal evidence and the underlying mechanisms are poorly understood.

We have recently found that genetic deletion of IGF binding protein 5a (Igfbp5a), a major PAPP-A substrate, results in reduced calcified bone mass in zebrafish even though *igfbp5a* is not expressed in skeletal tissues (*Liu et al., 2018*). In zebrafish embryos and larvae, *igfbp5a* is specifically expressed in a population of $Ca^{2+}$-transporting epithelial cells (ionocytes) located in the yolk sac (*Dai et al., 2014*; *Liu et al., 2017*). These ionocytes, known as NaR cells, are functionally similar to human intestinal epithelial cells. They play a key role in maintaining body $Ca^{2+}$ homeostasis by uptaking $Ca^{2+}$ from the surrounding habitat, (*Hwang, 2009*; *Lin and Hwang, 2016*). A hallmark of NaR cells and human intestinal epithelial cells is the expression of Trpv6/TRPV6, a constitutive calcium channel constituting the first and rate-limiting step in the transcellular $Ca^{2+}$ transport pathway (*Hoenderop et al., 2005*; *Pan et al., 2005*; *Dai et al., 2014*). Trpv6/TRPV6 also regulates NaR cell quiescence (*Xin et al., 2019*). NaR cells, normally non-dividing and quiescent, rapidly exit quiescence and re-enter the cell cycle in response to low $[Ca^{2+}]$ stress (*Dai et al., 2014*; *Liu et al., 2017*). This is thought to be an adaptive response, allowing animals to take up adequate $Ca^{2+}$ for maintaining body $Ca^{2+}$ homeostasis and survive under low $[Ca^{2+}]$ conditions (*Liu et al., 2018*). Interestingly, while no change was observed in NaR cells under normal $[Ca^{2+}]$ conditions, the lower $[Ca^{2+}]$ stress-induced adaptive NaR cell reactivation and proliferation were impaired in *igfbp5a-/-* fish (*Liu et al., 2018*). The mechanism underlying this $[Ca^{2+}]$-dependent phenotype is unclear.

Zebrafish have three genes belonging to the pappalysin family, including *papp-aa*, *papp-ab*, and *papp-a2* (*Kjaer-Sorensen et al., 2013*; *Kjaer-Sorensen et al., 2014*; *Wolman et al., 2015*). In this study, we show that among the three genes, *papp-aa* is highly expressed in NaR cells. Genetic deletion of *papp-aa* but not the paralogous *papp-ab*, resulted in a lack of calcified bone and abolished low $[Ca^{2+}]$ stress-induced NaR cell reactivation and proliferation. We provide several lines of independent evidence showing that Papp-aa-mediated Igfbp5a proteolysis acts as a $[Ca^{2+}]$-regulated switch linking local IGF signaling to epithelial cell proliferation and organismal $Ca^{2+}$ balance.

## Results

### Papp-aa is highly expressed in NaR cells and is indispensable in NaR cell reactivation and bone calcification

To date, published whole mount in situ hybridization data suggest that *papp-aa* mRNA is expressed in various neural tissues, *papp-ab* mRNA in developing myotomes and brain (*Kjaer-Sorensen et al., 2013*; *Wolman et al., 2015*; *Miller et al., 2018*; *Alassaf et al., 2019*), and *papp-a2* in the notochord and brain (*Kjaer-Sorensen et al., 2014*). Because NaR cells are located in the yolk sac epidermis, they are more sensitive to protease K treatment, a key step in the whole mount in situ hybridization procedure to permeabilize embryos. To test whether any of the pappalysin genes are expressed in NaR cells, we isolated NaR cells from *Tg(igfbp5a:GFP)* fish using FACS. *Tg(igfbp5a:GFP)* fish are a reporter fish line in which NaR cells are labeled by GFP expression (*Liu et al., 2017*). The mRNA levels of *papp-aa* in NaR cells were 2-fold higher than those of *papp-ab* and *papp-a2* (*Figure 1A*). Low $[Ca^{2+}]$ stress treatment had no effect on their mRNA levels (*Figure 1A*). We also compared the *papp-aa* mRNA levels in NaR cells with those non-GFP cells from the rest of the fish body. The level of *papp-aa* mRNA in NaR cells was approximately 10-fold greater (*Figure 1B*). In comparison, the *papp-ab* mRNA levels were similar between NaR cells and other cells (*Figure 1C*). Next, whole mount in situ hybridization was performed after optimizing the permeabilization condition. In agreement with previous reports (*Wolman et al., 2015*), strong *papp-aa* mRNA signal was detected in the brain (*Figure 1D*). Meanwhile, *papp-aa* mRNA signals were detected in cells in the yolk sac region beginning at three dpf (*Figure 1D*). Double color label staining showed that *papp-aa* mRNA was indeed expressed in NaR cells (*Figure 1E*).

Next, a blind test was carried out to investigate the possible function of Papp-aa in NaR cells. For this, progeny from *papp-aa+/-* fish (*Wolman et al., 2015*) intercrosses, which were a mixture of

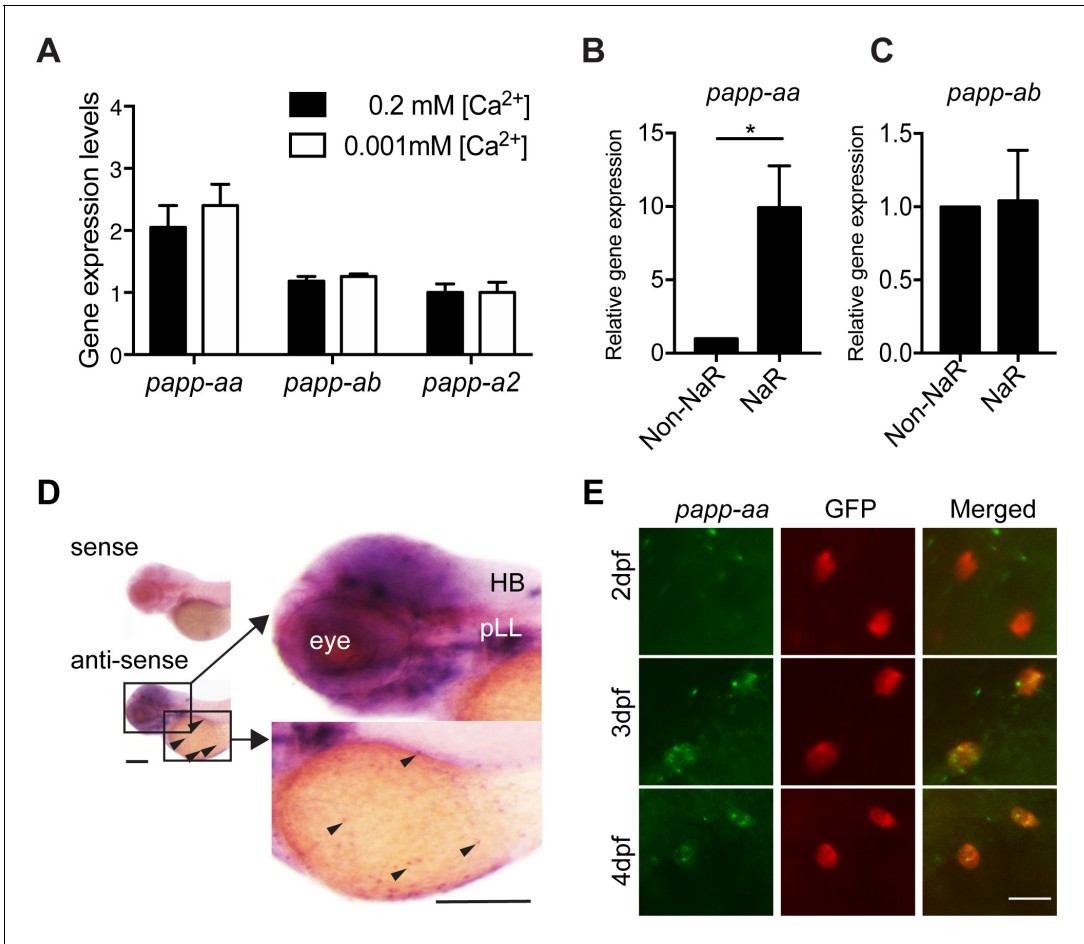

**Figure 1.** Papp-aa is highly expressed in NaR cells. (**A**) *Tg(igfbp5a:GFP)* fish were raised in E3 embryo medium to 3 days post fertilization (dpf) and transferred to embryo media containing the indicated [$Ca^{2+}$]. Eighteen hours later, NaR cells were isolated by FACS. The levels of *papp-aa*, *papp-ab*, and *papp-a2* mRNA were measured and shown. Data shown are Mean ± SEM, n = 4. (**B–C**) NaR cells and other cells in four dpf *Tg(igfbp5a:GFP)* larvae were separated by FACS. The levels of *papp-aa* (**B**) and *papp-ab* (**C**) mRNA were measured and shown. *, p<0.05 by unpaired two-tailed t test. n = 3. (**D**) Whole mount in situ hybridization analysis of *papp-aa* mRNA in three dpf larvae. HB, hindbrain. pLL, posterior lateral line ganglion. Arrowheads indicate *papp-aa* mRNA signal in the yolk sac region. A sense cRNA probe was used as a negative control. Scale bar = 0.2 mm. (**E**) *Tg(igfbp5a:GFP)* fish of the indicated stages were analyzed by double label staining. Scale bar = 20 µm.
The online version of this article includes the following source data for figure 1:

**Source data 1.** Excel spreadsheet containing quantitative data for *Figure 1*.

mutant, heterozygous, and wild type embryos, were raised in the standard E3 embryo medium until three dpf. The fish were transferred to normal or low [$Ca^{2+}$] embryo rearing solutions at three dpf and sampled at five dpf (*Figure 2A*). NaR cells were visualized by in situ hybridization using an *igfbp5a* cRNA probe and quantified. Larvae were subsequently genotyped individually (*Figure 2A*). Under the normal [$Ca^{2+}$] condition, NaR cell numbers between these three different genotype groups were similar (*Figure 2B and C*). Low [$Ca^{2+}$] stress treatment, however, resulted in a 4-fold, highly significant increase in NaR cell number in the wild-type and heterozygous fish groups. This increase was abolished in *papp-aa$^{-/-}$* mutant fish (*Figure 2B and C*). Similar results were obtained using *trpv6* mRNA as a NaR cell marker (*Figure 2—figure supplement 1A*). To test whether this action is specific to Papp-aa, we deleted the paralogous *papp-ab* gene using CRISPR-Cas9. Deletion of Papp-ab had no effect on low [$Ca^{2+}$] stress-induced NaR cell increase nor did it affect NaR cell number under the normal [$Ca^{2+}$] condition (*Figure 2—figure supplement 2*).

Next, we examined NaR cell reactivation and proliferation using the progeny of *papp-aa$^{+/-}$;Tg(igfbp5a:GFP)* fish intercrosses. While NaR cells in *papp-aa$^{+/+}$;Tg(igfbp5a:GFP)* larvae and *papp-aa$^{+/-}$;Tg(igfbp5a:GFP)* larvae divided robustly under low [$Ca^{2+}$] stress, NaR cell re-activation was

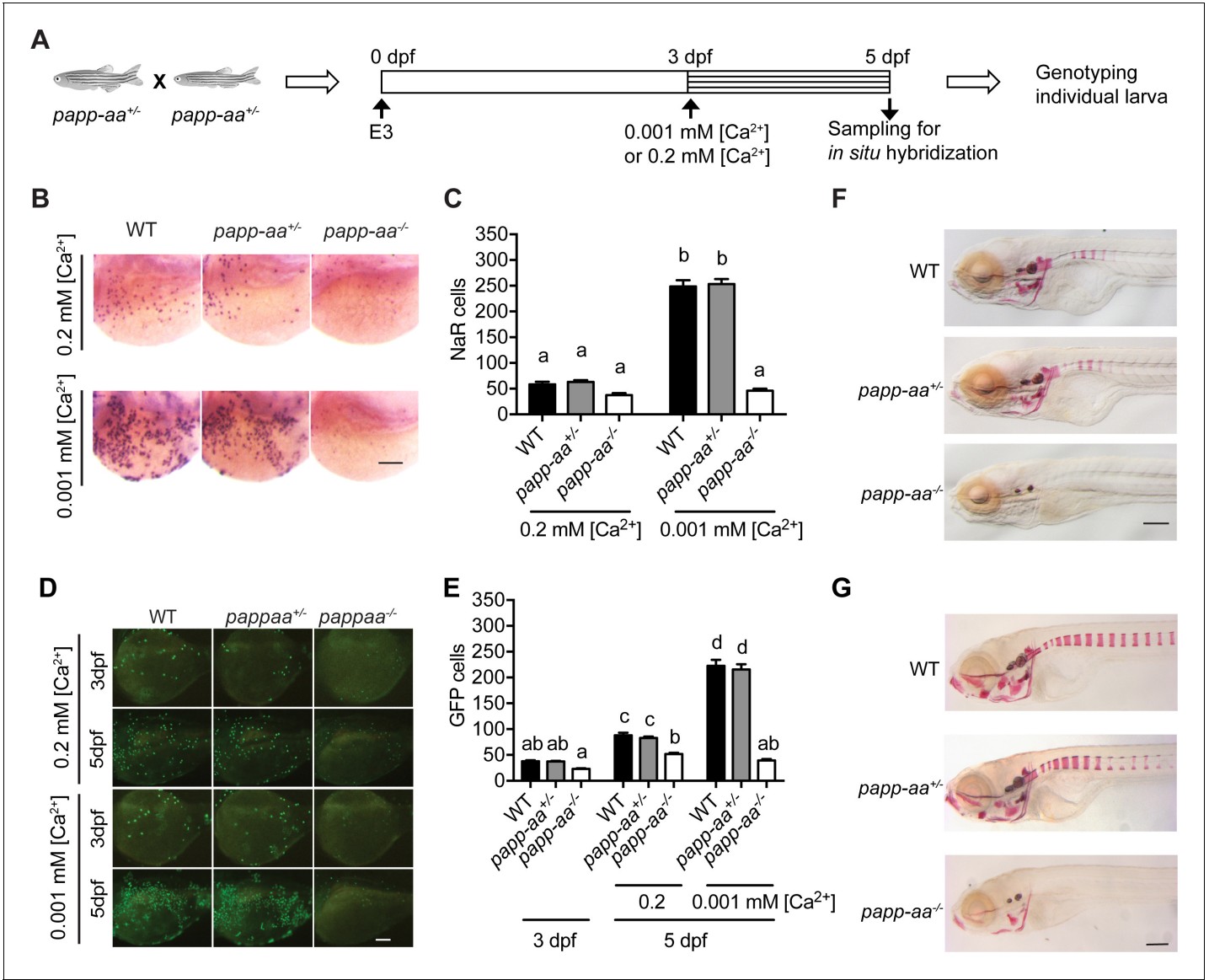

**Figure 2.** Genetic deletion of *papp-aa* impairs NaR cell reactivation and bone calcification. (A) Diagram of the experimental design. Progeny of *papp-aa*[+/-] fish intercrosses were raised in standard E3 embryo medium to three dpf. The progeny is a mixture of homo, hetero, and wild type embryos. They were transferred to the low [Ca$^{2+}$] (0.001 mM) or normal [Ca$^{2+}$] (0.2 mM) embryo medium at three dpf. Two days later, NaR cells in each fish were detected by *igfbp5a* mRNA expression and quantified. These fish were genotyped individually afterwards. (B–C) Progeny of *papp-aa*[+/-] intercrosses were treated as described in (A). Representative images are shown in (B) and quantified data in (C). Scale bar = 0.1 mm. n = 10 ~ 30 fish/group. In this and all subsequent figures, data shown are Mean ± SEM. Different letters indicate significant differences among groups by one-way ANOVA followed by Tukey's multiple comparison test (p<0.05). (D–E) Progeny of *papp-aa*[+/-];*Tg(igfbp5a:GFP)* fish intercrosses were raised in E3 medium to three dpf and transferred to the low [Ca$^{2+}$] (0.001 mM) or normal [Ca$^{2+}$] (0.2 mM) embryo medium. Two days later, the number of GFP-expressing NaR cells in each larva was quantified. The larvae were genotyped individually subsequently. Representative images are shown in (D) and quantified data in (E). Scale bar = 0.1 mm. n = 16 ~ 82 fish/group. (F–G) Fish of the indicated genotypes were raised in E3 embryo medium to 7 dpf and 12 dpf and stained with Alizarin Red. Scale bar = 0.2 mm.

The online version of this article includes the following source data and figure supplement(s) for figure 2:

**Source data 1.** Excel spreadsheet containing quantitative data for *Figure 2*.

**Figure supplement 1.** Genetic deletion of *papp-aa* impairs NaR cell proliferation, but has no effect on HR and NCC cells.

**Figure supplement 1—source data 1.** Excel spreadsheet containing quantitative data for *Figure 2—figure supplement 1*.

**Figure supplement 2.** Genetic deletion of *papp-ab* has no effect on NaR cell proliferation.

**Figure supplement 2—source data 1.** Excel spreadsheet containing quantitative data for *Figure 2—figure supplement 2*.

**Figure supplement 3.** Global and local effects of genetic deletion of *papp-aa*.

**Figure supplement 3—source data 1.** Excel spreadsheet containing quantitative data for *Figure 2—figure supplement 3*.

*Figure 2 continued on next page*

*Figure 2 continued*

**Figure supplement 4.** Genetic deletion of *papp-aa* impairs bone calcification.

**Figure supplement 4—source data 1.** Excel spreadsheet containing quantitative data for *Figure 2—figure supplement 4*.

abolished in *papp-aa$^{-/-}$;Tg(igfbp5a:GFP)* larvae (*Figure 2D and E*). This data suggest that Papp-aa plays a critical role in low [Ca$^{2+}$] stress-induced NaR cell reactivation and proliferation.

To test whether the action of Papp-aa is restricted to NaR cells, we examined H$^+$-ATPase-rich (HR) cells and Na$^+$/Cl$^-$ cotransporter (NCC) cells, two other types of ionocytes located in the yolk sac epidermis. HR and NCC cells are responsible for H$^+$ secretion/Na$^+$ uptake/NH$^+$ excretion and Na$^+$ uptake/Cl$^-$ uptake, respectively (*Hwang, 2009*). Deletion of Papp-aa did not change HR or NCC cell numbers (*Figure 2—figure supplement 1B and C*).

No difference was observed between *papp-aa$^{-/-}$* mutant fish and their siblings in terms of gross appearance, growth rate, or developmental speed (*Figure 2—figure supplement 3A–E*). We noted that NaR cells in wild-type *Tg(igfbp5a:GFP)* larvae had a narrow apical opening when kept in the normal [Ca$^{2+}$] medium (*Figure 2—figure supplement 3F*). This apical opening became markedly

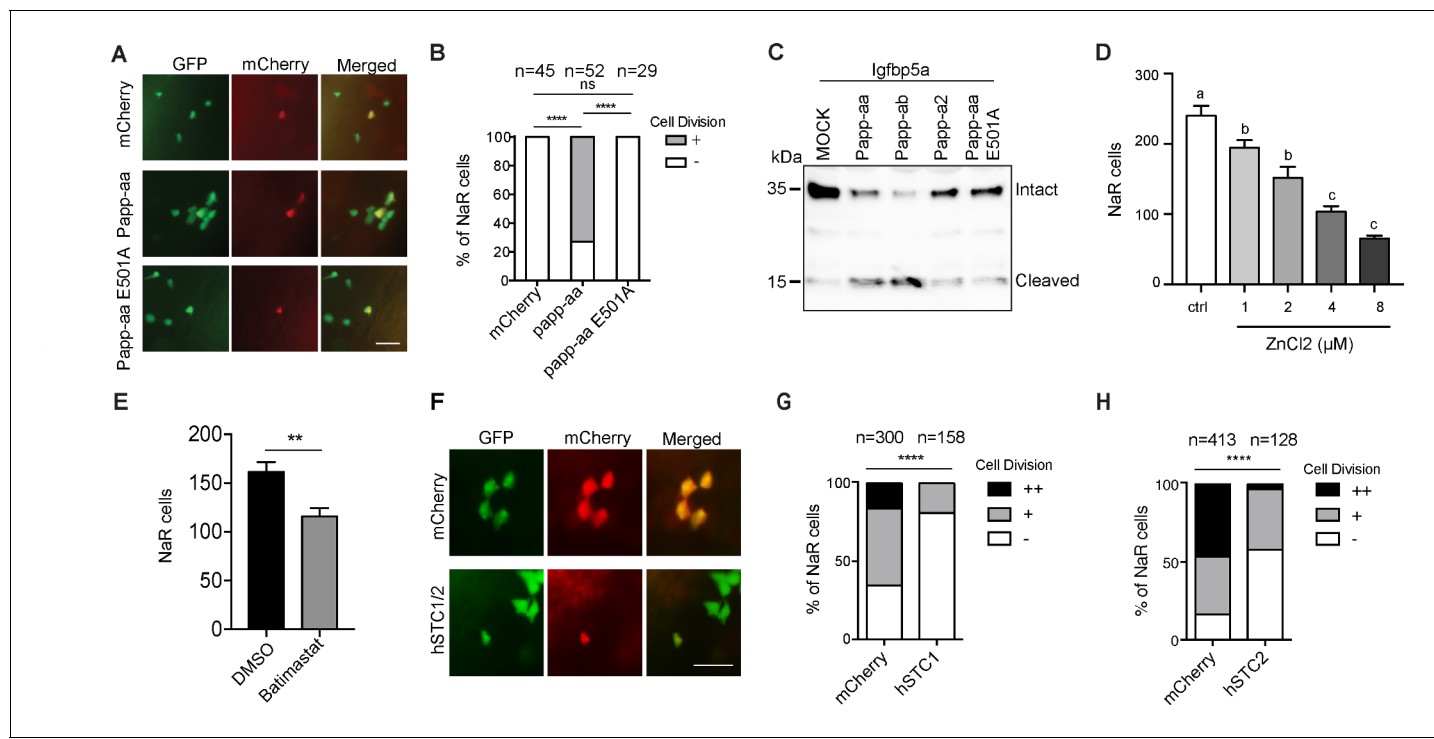

**Figure 3.** Papp-aa proteinase activity is in NaR cells critical. (**A–B**) Progeny of *papp-aa$^{+/-}$;Tg(igfbp5a:GFP)* intercrosses were injected with *BAC(igfbp5a:mCherry)* containing the indicated gene. They were subjected to the low [Ca$^{2+}$] stress test described in *Figure 2A*. Papp-aa-IRES-mCherry, Papp-aa E501A-IRES-mCherry, or mCherry expressing NaR cells were detected by GFP and mCherry expression. NaR cells expressing mCherry or Papp-aa-mCherry (yellow, double labeled by GFP and mCherry) were scored following a published scoring system (*Liu et al., 2018*). Representative images are shown in (**A**) and quantified data in (**B**). Scale bar = 50 µm. +, one cell division, -, no division. ****, p<0.0001 by Chi-square test. The total cell number is shown above the bar. (**C**) Conditioned media collected from HEK293 cells co-transfected with Igfbp5a and the indicated plasmid were analyzed by western blotting. Intact and cleaved Igfbp5a bands were indicated. (**D–E**) *Tg(igfbp5a:GFP)* fish were transferred to the low [Ca$^{2+}$] medium containing 0–8 µM ZnCl$_2$ (D) or 200 µM Batimastat at three dpf (E). After two days of treatment, NaR cells were quantified and shown. n = 18 ~ 25 fish/group. **, p<0.001 by unpaired two-tailed t test. (**F–H**) *Tg(igfbp5a:GFP)* embryos were injected with *BAC(igfbp5a:mCherry, BAC(igfbp5a:hSTC1-IRES-mCherry)* (G) or *BAC(igfbp5a:hSTC2-IRES-mCherry)* (H). They were raised and subjected to the low [Ca$^{2+}$] stress test described in *Figure 2A*. NaR cells expressing mCherry or human STC (yellow, double labeled by GFP and mCherry) were scored following a published scoring system (*Liu et al., 2018*). Representative images are shown in (**F**) and quantified results in (**G and H**). ++, two cell division, +, one cell division, -, no division during the experiment. ****, p<0.0001, Chi-square test. Total cell number is shown above the bar.

The online version of this article includes the following source data for figure 3:

**Source data 1.** Excel spreadsheet containing quantitative data for *Figure 3*.

enlarged under low [Ca$^{2+}$] stress treatment (*Figure 2—figure supplement 3F*). Although the functional significance of this morphological change is unclear, this low [Ca$^{2+}$] stress-induced morphological change in NaR was absent in *papp-aa$^{-/-}$* mutant fish (*Figure 2—figure supplement 3F*). Alizarin red staining analysis indicated a complete lack of calcified bone in *papp-aa$^{-/-}$* fish at seven dpf (*Figure 2F*). At 10 and 12 dpf, some components of the craniofacial skeleton in *papp-aa$^{-/-}$* fish were calcified, albeit at very low levels (*Figure 2G*; *Figure 2—figure supplement 4*). However, none of the vertebral columns in *papp-aa$^{-/-}$* fish were stained by Alizarin red even at 10 and 12 dpf, suggesting the mutant fish suffered from Ca$^{2+}$ deficiency (*Figure 2—figure supplement 4*). In agreement with a previous report (*Wolman et al., 2015*), *papp-aa$^{-/-}$* mutant fish died around 2 weeks when kept under normal [Ca$^{2+}$]. Together, these results suggest that Papp-aa plays an indispensable role in regulating bone calcification.

## Endogenous Papp-aa proteinase activity in NaR cells is critical

To determine whether Papp-aa expression is sufficient in inducing NaR cell reactivation and proliferation, Papp-aa-IRES-mCherry was randomly expressed in NaR cells in *papp-aa$^{-/-}$*; *Tg(igfbp5a:GFP)* fish using a Tol2 transposon and BAC-mediated genetic mosaic assay (*Liu et al., 2018*). Re-introduction of Papp-aa restored low [Ca$^{2+}$] stress-induced NaR cell reactivation (*Figure 3A and B*). Previous studies suggested that pappalysins can affect zebrafish development in both proteinase activity-dependent as well as independent manners (*Kjaer-Sorensen et al., 2013*; *Kjaer-Sorensen et al., 2014*). To test whether this action of Papp-aa requires its proteinase activity, a conserved glutamate residue (E501) in the active site known to be critical for human PAPP-A proteinase activity (*Boldt et al., 2001*) was changed to alanine. As expected, while Papp-aa cleaved Igfbp5a, Papp-aa E501A had little proteolytic activity (*Figure 3C*). When the E501A mutant was expressed in NaR cells in *papp-aa$^{-/-}$*; *Tg(igfbp5a:GFP)* using the genetic mosaic assay, it had no effect in rescuing NaR cell reactivation (*Figure 3A and B*). Although the Tol2 transposon BAC-mediated genetic mosaic assay is useful in testing the sufficiency of Papp-aa and its mode of action, this approach cannot be used to rescue the lethality phenotype because it only targets the expression of Papp-aa transiently and to a small subset of NaR cells in a mosaic pattern.

To test the role of Papp-aa further, *Tg(igfbp5a:GFP)* embryos were treated with ZnCl$_2$ or batimastat, two metzincin metalloproteinase inhibitors (*Tallant et al., 2006*). Both compounds inhibited low [Ca$^{2+}$] stress-induced NaR cell proliferation (*Figure 3D and E*). Next, we targeted expression of human stanniocalcin (STC) 1 and 2 in NaR cells, respectively. STC2 has been shown to be able to inhibit PAPP-A activity by forming a covalent complex with PAPP-A, while STC1 inhibits PAPP-A activity by non-covalent protein-protein interaction (*Jepsen et al., 2015*; *Kløverpris et al., 2015*). When STC1 was specifically expressed in a subset of NaR cells using the genetic mosaic assay, it inhibited low [Ca$^{2+}$] stress-induced NaR cell proliferation, while mCherry expression had no such effect (*Figure 3F and G*). Likewise, targeted expression of STC2 also inhibited low [Ca$^{2+}$] stress-induced NaR cell proliferation (*Figure 3H*). These results suggest that endogenous Papp-aa proteinase activity in NaR cells is required for NaR cell reactivation and proliferation in response to low [Ca$^{2+}$] stress.

## Papp-aa-mediated Igfbp5 proteolysis in NaR cells is critical

Human PAPP-A is a specific IGFBP4/5 proteinase (*Oxvig, 2015*). Among the three zebrafish pappalysin family members, Papp-ab has been shown to cleave human IGFBP4 and IGFBP5 but not the other four IGFBPs (*Kjaer-Sorensen et al., 2013*). Zebrafish Papp-a2 can cleave human IGFBP3 and IGFBP5, but not other IGFBPs (*Kjaer-Sorensen et al., 2014*). The substrate specificity of zebrafish Papp-aa has not been reported. We therefore performed an in vitro proteinase assay using purified human IGFBPs and conditioned media collected from HEK293 cells transfected with Papp-aa cDNA or empty vector (mock medium). As shown in *Figure 4A*, Papp-aa containing medium efficiently cleaved IGFBP5 and IGFBP4, whereas it did not cleave IGFBP1, 2, 3 and 6 (*Figure 4A*).

The zebrafish genome has no *igfbp4* gene, but contains two paralogous *igfbp5* genes (*Allard and Duan, 2018*). To show that zebrafish Papp-aa cleaves Igfbp5a and to clarify the relationship among multiple pappalysins and Igfbp5s, in vitro proteinase assays were carried out using homologous proteins. Both zebrafish Papp-aa and Papp-ab cleaved Igfbp5a, whereas Papp-a2 had little activity (*Figure 3C*). We noted a substantial level of Igfbp5a degradation by mock media,

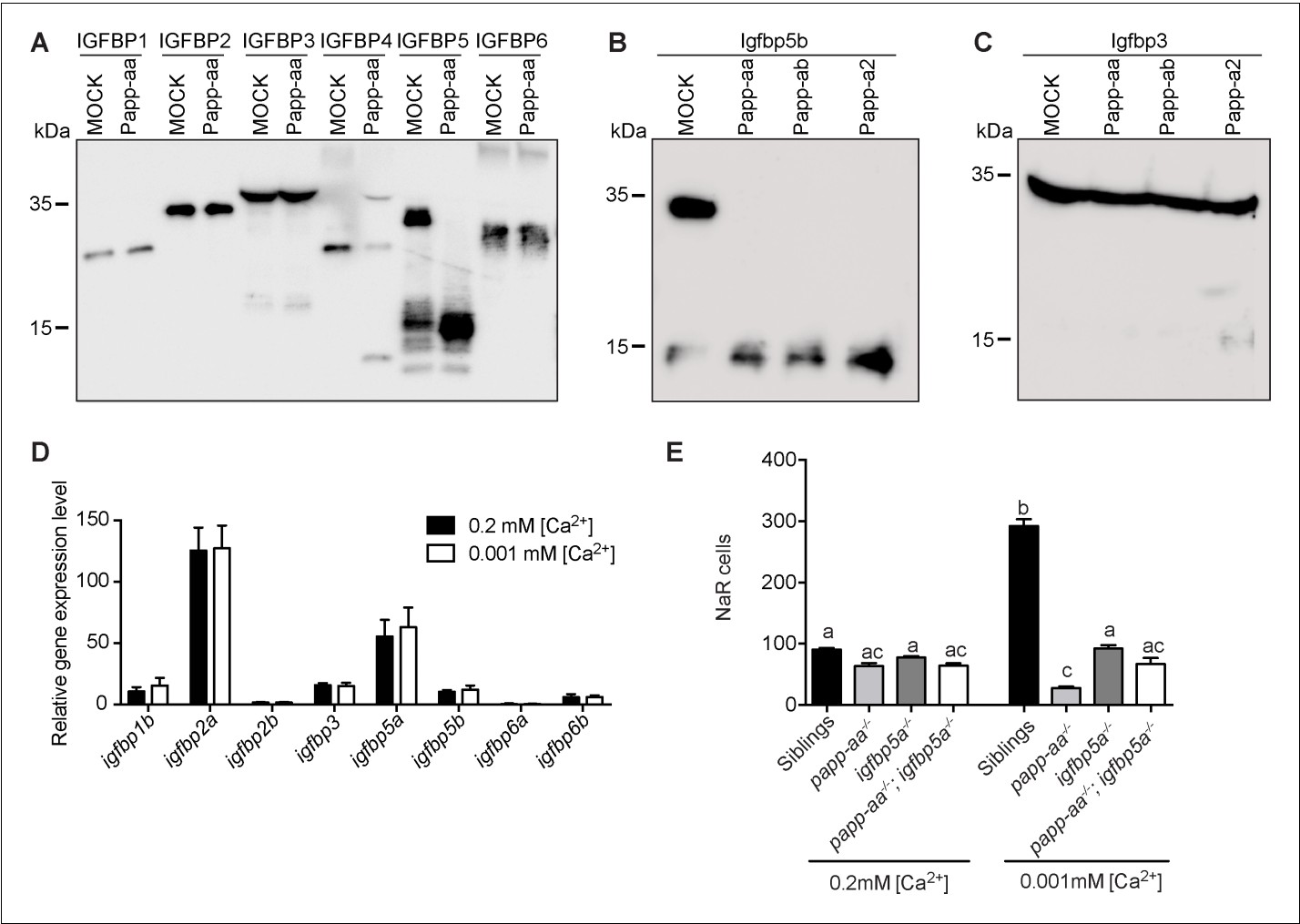

**Figure 4.** Papp-aa is an Igfbp5 proteinase and Papp-aa-mediated Igfbp5 proteolysis in NaR cells is critical. (**A**) Purified human IGFBP1, 2, 3, 4, 5, and six was incubated for 2 hr at 28°C with conditioned media collected from HEK293 cells transfected with zebrafish Papp-aa and from Mock transfected cells. For the IGFBP4 group, 100 nM IGF1 was added prior to the initiation of the proteinase assay. The protease reaction was stopped and analyzed by western blotting. (**B**) Conditioned media collected from HEK293 cells co-transfected with Igfbp5b and the indicated plasmid were analyzed by western blotting. Intact and cleaved Igfbp5b bands were indicated. (**C**) Conditioned media collected from HEK293 cells co-transfected with Igfbp3 and the indicated plasmid were analyzed by western blotting. Intact and cleaved Igfbp3 bands were indicated. (**D**) *Tg(igfbp5a:GFP)* fish were raised in E3 embryo medium to 3 days post fertilization (dpf) and transferred to embryo media containing the indicated [Ca$^{2+}$]. Eighteen hours later, NaR cells were isolated by FACS. The mRNA levels of the indicated *igfbp* genes were measured and shown. Data shown are Mean ± SEM, n = 4. (**E**) Progenies resulted from *papp-aa$^{+/-}$; igfbp5a$^{+/-}$* and *papp-aa$^{+/-}$; igfbp5a$^{-/-}$* intercrosses were raised in E3 embryo medium to three dpf and transferred to the normal [Ca$^{2+}$] (0.2 mM) or low [Ca$^{2+}$] (0.001 mM) embryo medium. At five dpf, NaR cells in each fish were labeled by *igfbp5a* mRNA expression and quantified. Each fish was genotyped individually afterward. Quantified data are shown. n = 18 ~ 63 fish/group.

The online version of this article includes the following source data and figure supplement(s) for figure 4:

**Source data 1.** Excel spreadsheet containing quantitative data for *Figure 4*.

**Figure supplement 1.** Conservation of Papp-aa cleavage site in Igfbp5a and 5b.

indicating the presence of other Igfbp5a degrading proteases in HEK293 cells. In the case of Igfbp5b, it was cleaved by Papp-aa, Papp-ab, and Papp-a2 (*Figure 4B*). K128 in human IGFBP5 is critical for PAPP-A-mediated cleavage (*Laursen et al., 2001*; *Laursen et al., 2002*), and this residue is present in zebrafish Igfbp5a (K147 or K148) and Igfbp5b (K144 or K145) (*Figure 4—figure supplement 1A*). Indeed, changing Igfbp5a K148 to alanine reduced Papp-aa-mediated cleavage, while mutating the neighboring K147 had little effect (*Figure 4—figure supplement 1B*). Likewise, changing Igfbp5b K145 to alanine reduced Papp-aa-mediated Igfbp5b cleavage, while K144A mutation had little effect (*Figure 4—figure supplement 1C*). These results suggest that Papp-aa and Papp-

ab, but not Papp-a2, can cleave Igfbp5a and the cleavage site appears to be conserved. We also tested Igfbp3, which is most closely related to Igfbp5 among all the Igfbps (*Duan and Allard, 2020*). Papp-aa and Papp-ab did not cleave Igfbp3 and Papp-a2 had very low activity (*Figure 4C*).

Gene expression analysis results showed that *Igfbp5a* and *Igfbp2a* are highly expressed in NaR cells (*Figure 4D*). The levels of *igfbp5b*, *igfbp1b*, *igfbp3*, and *igfbp6b* mRNA in NaR cells are much lower, and *igfbp2b* and *igfbp6a* are barely detectable (*Figure 4D*). Low [Ca$^{2+}$] stress treatment did not affect any of these *igfbp* mRNA levels (*Figure 4D*). As mentioned above, *papp-aa* but not *papp-ab* is highly expressed in NaR cells (*Figure 1*). These gene expression data together with the protein-ase assay results suggest that Igfbp5a is likely the primary Papp-aa substrate in NaR cells.

To further test the functional importance of Papp-aa-mediated Igfbp5a proteolytic cleavage in vivo, we crossed *papp-aa$^{-/-}$* and *igfbp5a$^{-/-}$* lines and generated double mutant embryos. Again, the phenotypes of the progeny were analyzed in a blind fashion followed by individual genotyping. As shown in *Figure 4E*, under normal [Ca$^{2+}$] conditions, NaR cell numbers in *papp-aa$^{-/-}$* fish, *igfbp5a$^{-/-}$* fish, *papp-aa$^{-/-}$;igfbp5a$^{-/-}$* double mutant fish, and siblings were similar and no significant difference was observed (*Figure 4E*). While the low [Ca$^{2+}$] stress treatment resulted in a significant increase in NaR cell number in the siblings, this increase was abolished in *papp-aa$^{-/-}$* fish, *igfbp5a$^{-/-}$* fish, and *papp-aa$^{-/-}$;igfbp5a$^{-/-}$* double mutant fish (*Figure 4E*). These genetic results suggest that Papp-aa-mediated Igfbp5a proteolysis is critical for NaR cell reactivation.

## Papp-aa promotes IGF signaling and NaR cell proliferation by increasing bioavailable IGFs

As previously reported (*Dai et al., 2014*; *Liu et al., 2018*), low [Ca$^{2+}$] stress treatment activates Akt and Tor activity in wild-type fish NaR cells (*Figure 5A and B*; *Figure 5—figure supplement 1A*). This induction was impaired in *papp-aa$^{-/-}$* fish but not in the heterozygous fish (*Figure 5A and B*; *Figure 5—figure supplement 1A*). The low [Ca$^{2+}$] stress-induced Akt signaling activity was similarly impaired in *papp-aa$^{-/-}$* single mutant and *papp-aa$^{-/-}$; igfbp5a$^{-/-}$* double mutant fish (*Figure 5—figure supplement 1*). Treatment of wild-type zebrafish larvae with ZnCl$_2$ and batimastat also inhibited the low [Ca$^{2+}$] stress-induced Akt and Tor activity (*Figure 5C and D*).

The above results indicate that Papp-aa promotes NaR cell reactivation via the Akt-Tor signaling pathway. If this is correct, then constitutive activation of this signaling pathway should rescue NaR cell reactivation in *papp-aa$^{-/-}$* fish. Indeed, genetic mosaic assays showed that targeted expression of myr-Akt, a constitutively active Akt (*Kohn et al., 1996*), restored NaR cell reactivation in *papp-aa$^{-/-}$* fish under to low [Ca$^{2+}$] stress (*Figure 5E and F*).

Because Papp-aa increases IGF signaling in NaR cells only under low [Ca$^{2+}$] stress, we speculated that Papp-aa activity is suppressed and IGF ligands are sequestered in the Igfbp5a/IGF complex under normal [Ca$^{2+}$] conditions. If this is correct, then pharmacological disruption of the Igfbp5a/IGF complex should activate IGF signaling and increase NaR cell reactivation under normal [Ca$^{2+}$] conditions. To test this idea we used NBI-31772, an aptamer that can displace and release IGF from the IGF/IGFBP complex (*Chen et al., 2001*). NBI-31772 treatment promoted NaR cell reactivation under normal [Ca$^{2+}$] in a concentration-dependent manner (*Figure 6A and B*). To test whether this effect of NBI-31772 is indeed mediated via IGF signaling, we co-treated zebrafish larvae with NBI-31772 and BMS-754807, an IGF1 receptor inhibitor and measured the levels of phospho-Akt and phospho-pS6 activity in NaR cells. While NBI-31772 treatment increased Akt and Tor signaling activity, these increases were abolished by BMS-754807 (*Figure 6C and D*). Moreover, the effect of NBI-31772 in reactivating NaR cells was abolished by adding BMS-754807, PI3 kinase inhibitor wortmannin, Akt inhibitor MK2260, and Tor inhibitor rapamycin (*Figure 6E*; *Figure 6—figure supplement 1B*). Mito-gen-activated kinase (MAPK) pathway is another major signaling pathway downstream of the IGF1 receptor (*Duan et al., 2010*). Immunostaining analysis results showed that NBI-31772 treatment did not change the phospho-Erk levels (*Figure 6—figure supplement 1A*). Co-treatment with the MEK inhibitor U0260 did not affect NBI-31772 treatment-induced NaR cell proliferation (*Figure 6—figure supplement 1B*), indicating MAPK signaling is not involved.

If the limiting step under normal [Ca$^{2+}$] is indeed the lack of bioavailable IGFs, then administering free IGF1 should stimulate NaR cell reactivation. This reasoning was tested by treating *Tg(igfbp5a: GFP)* embryos with free human IGF1. As shown in *Figure 6F*, IGF1 treatment significantly increased NaR cell reactivation and proliferation under normal [Ca$^{2+}$]. Likewise, fish (salmon) IGF1 treatment

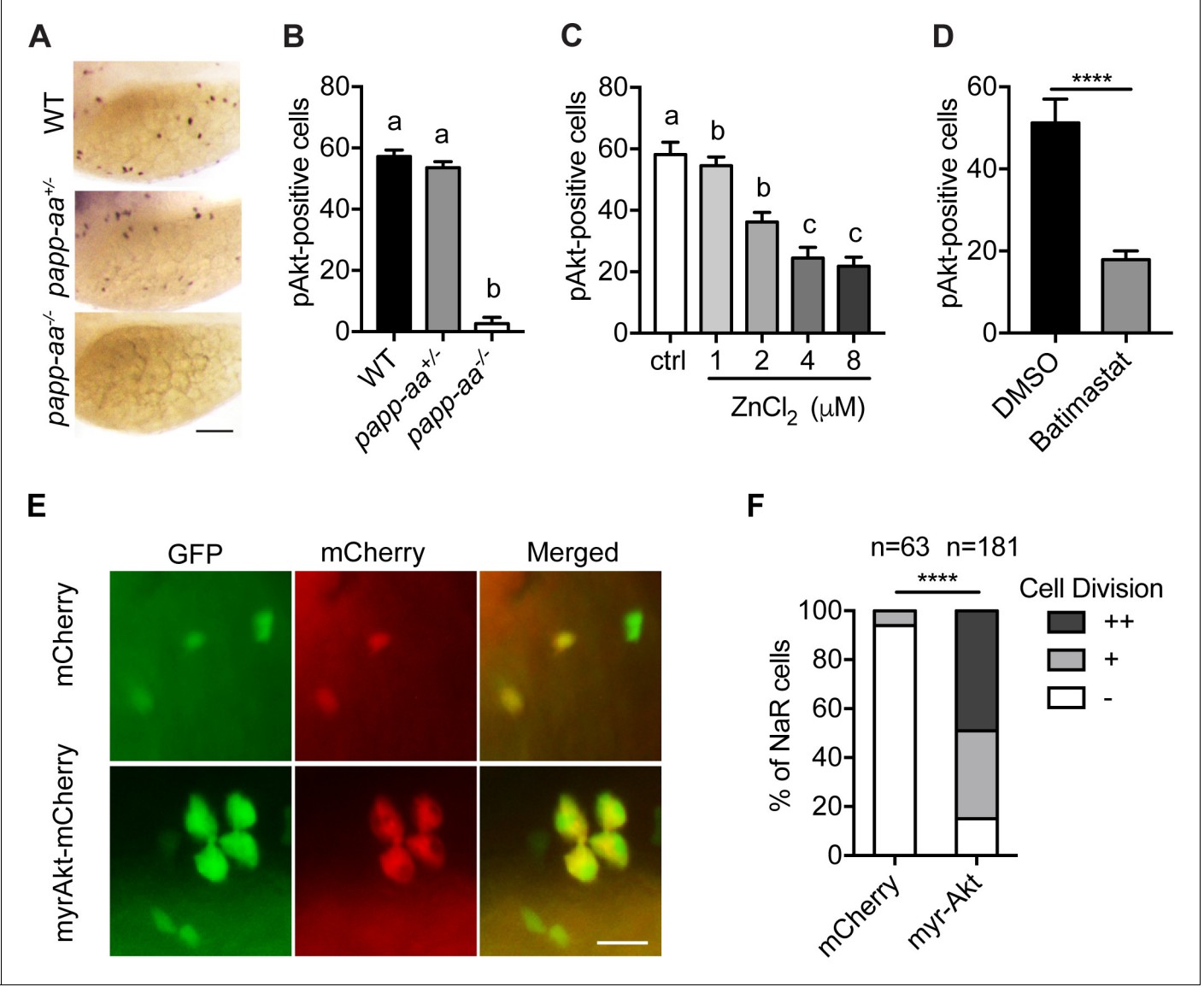

**Figure 5.** Papp-aa acts by regulating IGF-Akt-Tor signaling in NaR cells. (A–B) Zebrafish embryos of the indicated genotypes were transferred to the low [$Ca^{2+}$] medium at three dpf. One day later, they were fixed and stained for phospho-Akt. Representative images are shown in (A) and quantitative results in (B). Scale bar = 0.1 mm. n = 25 ~ 58 fish/group. (C–D) Tg(igfbp5a:GFP) fish were transferred to the low [$Ca^{2+}$] medium 0–8 μM ZnCl$_2$ (C) or 200 μM batimastat (D) at three dpf. After one day treatment, they were analyzed by immunostaining for phospho-Akt. n = 18 ~ 24 fish/group. ****, p<0.0001, unpaired two-tailed t test. (E and F) Progeny of papp-aa$^{+/-}$;Tg(igfbp5a:GFP) intercrosses were injected with BAC(igfbp5a:mCherry) or BAC (igfbp5a:myr-Akt-mCherry). They were raised and subjected to the low [$Ca^{2+}$] stress test described in *Figure 2A*. NaR cells expressing mCherry or myr-Akt were scored as described in *Figure 3*. Representative images are shown in (E) and quantified data in (F). ++, two cell division, +, one cell division, -, no division during the experiment. Scale bar = 20 μm. ****, p<0.0001 by Chi-square test. Total number of cells is shown above the bar.

The online version of this article includes the following source data and figure supplement(s) for figure 5:

**Source data 1.** Excel spreadsheet containing quantitative data for *Figure 5*.

**Figure supplement 1.** Papp-aa acts via Igfbp5a and IGF signaling in NaR cells.

**Figure supplement 1—source data 1.** Excel spreadsheet containing quantitative data for *Figure 5—figure supplement 1*.

also increased NaR cell reactivation and proliferation under normal [$Ca^{2+}$] conditions (*Figure 6—figure supplement 2*).

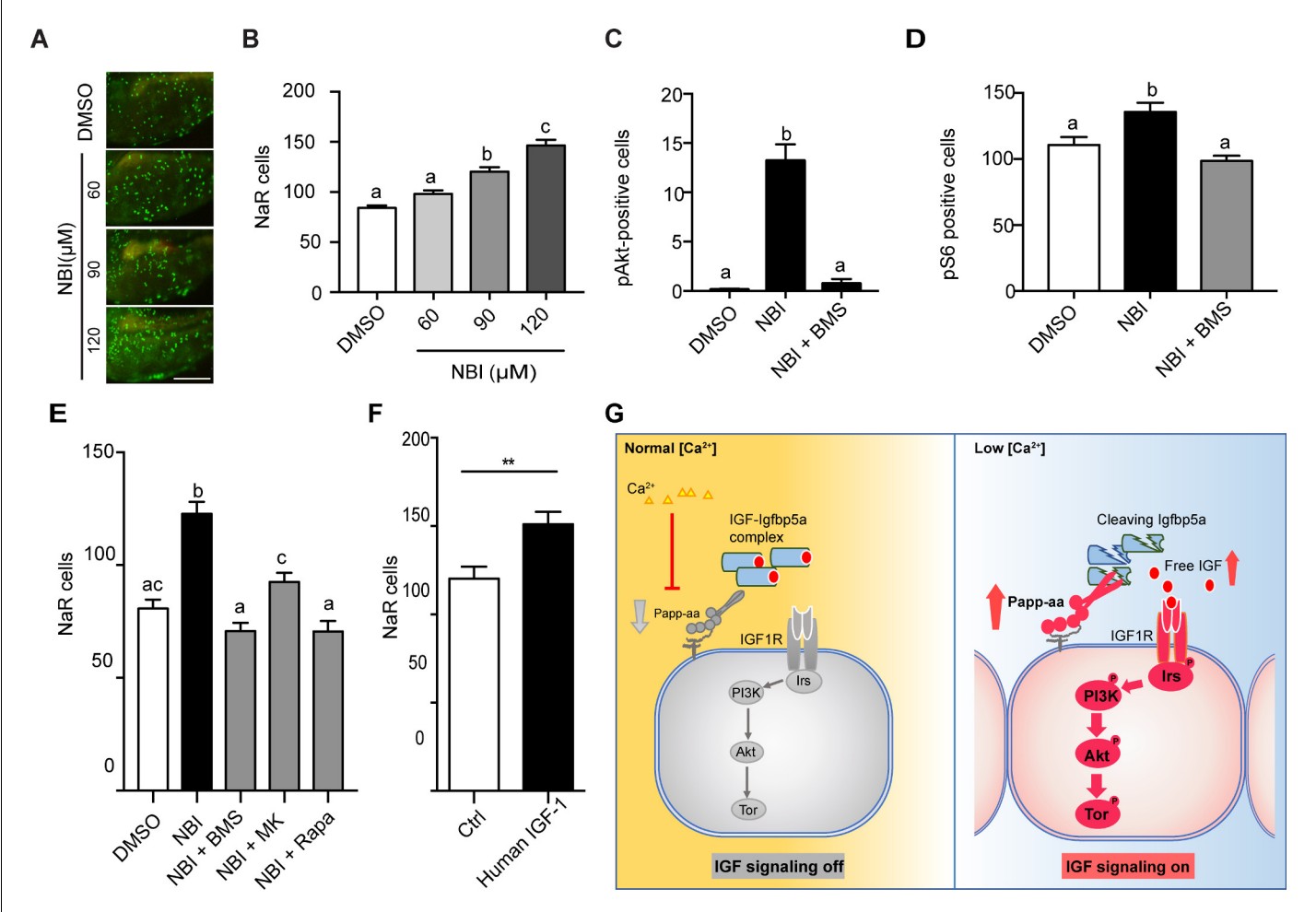

**Figure 6.** Disruption the IGF/Igfbp complex activates IGF-Akt-Tor signaling and promotes NaR cell reactivation. (A–B) *Tg(igfbp5a:GFP)* fish were transferred to normal [Ca²⁺] medium containing the indicated doses of NBI-31772 at three dpf. Two days later, NaR cells were quantified. Representative images are shown in (A) and quantified data in (B). Scale bar = 0.2 mm. n = 25 ~ 27 fish/group. (C) Wild-type fish were treated with 90 μM NBI-31772 with or without 0.3 μM BMS-754807 from 3 to 4 dpf. The number of cells positive for phosphorylated Akt staining were quantified and shown. n = 19 ~ 23 fish/group. (D) Larvae treated as described in (C) were stained for phosph-S6 and quantified. n = 14 ~ 15 fish/group. (E) *Tg(igfbp5a: GFP)* fish were treated with NBI-31772 (90 μM) together with BMS-754807 (0.3 μM), MK2206 (8 μM), or Rapamycin (5 μM) from 3 to 5 dpf. NaR cell number was quantified and shown. n = 10 ~ 24 fish/group. (F) *Tg(igfbp5a:GFP)* fish were treated with human IGF1 (150 ng/ml) in E3 embryo medium from 3 to 5 dpf. NaR cells were quantified and shown. n = 35 ~ 36 fish/group. **p<0.01, unpaired t-test. (G) Proposed model of Papp-aa function as a [Ca²⁺]-regulated molecular switch of IGF signaling in epithelial cells. Left panel: under normal [Ca²⁺] conditions, Papp-aa proteolysis activity is suppressed. Igfbp5a is intact and it inhibits IGF signaling by binding to IGFs and prevents their binding to the IGF1 receptor. The IGF- PI3 kinase-Akt-Tor signaling is inhibited in NaR cells. Right panel: under low [Ca²⁺] conditions, Papp-aa activity is increased. This increases Igfbp5a proteolytic cleavage and releases IGFs from the Igfbp5a/IGF complex. Bioavailable IGFs binds to IGF1 receptor and activates PI3 kinase-Akt-Tor signaling in NaR cells and promotes their reactivation and proliferate.

The online version of this article includes the following source data and figure supplement(s) for figure 6:

**Source data 1.** Excel spreadsheet containing quantitative data for *Figure 6*.
**Figure supplement 1.** Disruption the IGF/Igfbp complex has no effect on pErk activity and Mek/Erk signaling is not critical.
**Figure supplement 1—source data 1.** Excel spreadsheet containing quantitative data for *Figure 6—figure supplement 1*.
**Figure supplement 2.** Treatment with fish IGF1 increases NaR cell reactivation and proliferation.
**Figure supplement 2—source data 1.** Excel spreadsheet containing quantitative data for *Figure 6—figure supplement 2*.

## Discussion

In this study, we uncovered a novel mechanism regulating calcium homeostasis and bone calcification using the zebrafish model. Zebrafish have been used with growing success to model various pathological defects affecting human skeletal tissues (*Laizé et al., 2014*). Zebrafish shares many similarities in bone biology with mammals, including bone formation, resorption, mineralization, and extracellular matrix maintenance (*Laizé et al., 2014*). Unlike mammals, however, zebrafish bone is acellular and osteoclasts are mononucleated (*Laizé et al., 2014*). Calcium, an important bone component, is an essential ion and plays important roles in development and adult physiology. Calcium homeostasis is regulated by many hormonal circuits. Many hormones known to be critical to mammalian bone mineralization, including Vitamin D3, parathyroid hormone, calcitonin were found to be important in zebrafish (*Lin and Hwang, 2016*). There are also examples of hormones first discovered in fish. Stanniocalcin (Stc), for example, was initially discovered in bony fish as a hypocalcemic hormone (*Pang, 1973*). It was later found that humans and other mammals have two STC genes (*Yeung et al., 2012*). Recent studies suggest that human STC1 and STC2 are potent PAPP-A and A2 inhibitors in vitro (*Jepsen et al., 2015*; *Kløverpris et al., 2015*). In this study, we provide genetic, biochemical, and pharmacological evidence showing that Papp-aa and IGF signaling regulates bone calcification by promoting $Ca^{2+}$-transporting epithelial cell reactivation and proliferation. Mechanistically, Papp-aa acts by cleaving Igfbp5a and activating IGF1 receptor-mediated Akt and Tor signaling in NaR cells in response to low $[Ca^{2+}]$ stress.

Using an optimized in situ hybridization protocol and qPCR analysis of FACS isolated NaR cells, we found that Papp-aa is highly expressed in NaR cells. This was further confirmed by double color in situ hybridization. Our genetic analysis results showed that deletion of Papp-aa, but not the closely related Papp-ab, abolished low $[Ca^{2+}]$ stress-induced NaR cell quiescence-proliferation transition. There was a complete lack of calcified bone in *papp-aa$^{-/-}$* fish at seven dpf. Although low levels of calcification were detected in some components of the craniofacial skeleton in *papp-aa$^{-/-}$* fish at 10 and 12 dpf, the mutant fish had no calcified vertebrate in these late stages. The defective bone calcification and impaired NaR cell reactivation suggest that *papp-aa$^{-/-}$* fish suffered and died prematurely from $Ca^{2+}$ deficiency, likely due to insufficient epithelial $Ca^{2+}$ uptake. The action of Papp-aa on epithelial cells requires its proteinase activity. Reintroduction of wild-type Papp-aa, but not a proteolytically inactivated mutant, reactivated NaR cells in the *papp-aa$^{-/-}$* mutant fish. Likewise, treatment of wild-type fish with two distinct metalloproteinase inhibitors impaired NaR cell reactivation. When STC1 and STC2 was specifically expressed in NaR cells separately, they both blocked low $[Ca^{2+}]$ stress-induced NaR cell quiescence-proliferation transition, indicating the importance of local endogenous Papp-aa in NaR cells.

We provide several lines of independent evidence showing that Papp-aa regulates NaR cell quiescence-proliferation transition via proteolytically cleaving locally expressed Igfbp5a and activating IGF signaling. We have previously reported that Igfbp5a is specifically expressed in NaR cells (*Dai et al., 2014*; *Liu et al., 2017*). In this study, we found that in addition to Igfbp5a, Igfbp2a is also highly expressed in NaR cells. The expression levels of other Igfbps including Igfbp5b are low in NaR cells. Our biochemical analysis showed that Papp-aa efficiently cleaved human IGFBP4 and IGFBP5 but not IGFBP2 and other IGFBPs. Since the zebrafish genome does not have an igfbp4 gene, we postulated that Igfbp5a is the main Papp-aa substrate in NaR cells. We found that while Papp-aa cleaved Igfbp5a, Papp-a2 had no such activity. Although Papp-ab can also cleave Igfbp5a, its expression in NaR cells is relatively low. Another line of evidence came from *papp-aa$^{-/-}$* fish. The impaired NaR cell reactivation in *papp-aa$^{-/-}$* fish was accompanied with reduced IGF1 receptor-mediated Akt-Tor signaling activity. Likewise, inhibition of Papp-aa proteolytic activity resulted in reduced Akt-Tor activity in NaR cells. Importantly, targeted expression of a constitutively active Akt in *papp-aa$^{-/-}$* NaR cells was sufficient to rescue their reactivation and proliferation, supporting the notion that Akt mediates Papp-aa action in NaR cells. Akt is involved in signal transduction of a number of growth factors and hormones. However, the fact that addition of human or fish IGF-1 was sufficient to activate NaR cell proliferation strongly argues that IGF signaling is critical.

A most interesting finding made in this study is that Papp-aa-mediated Igfbp5a proteolysis in NaR cells is suppressed under normal $[Ca^{2+}]$ conditions. This conclusion is supported by the fact that genetic deletion of Papp-aa had no effect on Akt-Tor signaling or NaR cell proliferation when fish were kept under normal $[Ca^{2+}]$ conditions, while it abolished IGF signaling and NaR cell reactivation

and proliferation under low $[Ca^{2+}]$ stress. Pharmacological inhibition or targeted expression of STC1/2 affected IGF signaling in NaR cells only under low $[Ca^{2+}]$. Furthermore, pharmacological disruption of the Igfbp5/IGF complex by NBI-31772 treatment was sufficient to increase Akt and Tor activity and promote NaR cell reactivation under normal $[Ca^{2+}]$ conditions. This effect of NBI-31772 is clearly mediated by IGF signaling because 1) NBI-31772 increased Akt and Tor signaling activity in an IGF1 receptor-dependent manner, and 2) NBI-31772 treatment-induced NaR cell proliferation was abolished by co-treatment with inhibitors blocking the IGF1 receptor, Akt, PI3 kinase, and Tor activity. NBI-31772 not only binds to IGFBP5, but also other IGFBPs. Indeed, our RNA expression analysis results showed that not only Igfbp5a, but also Igfbp2a is highly expressed in NaR cells. While the levels of *igfbp1b*, *igfbp3*, *igfbp5b*, and *igfbp6b* mRNA in NaR cells are low, they are nonetheless detectable. Therefore, we cannot exclude the possibility that NBI-31772 may also affect IGFs bound to other Igfbps. However, regardless of whether Papp-aa acts via Igfbp5a alone or together with other Igfbps, the NBI-31772 results clearly indicate that latent IGF is present in/near NaR cells under normal $[Ca^{2+}]$, suggesting the limiting step may be the release of IGFs from the Igfbp/IGF complexes. The zebrafish genome contains 4 IGF genes (*Zou et al., 2009*). Currently, there are no antibodies available against these zebrafish IGF ligands, making it impossible to measure IGF release after NBI-31772 treatment. To circumvent this problem, we treated zebrafish embryos with free IGF1 following a recent paper (*Alassaf et al., 2019*). Addition of either human or fish IGF-1 increased NaR cell proliferation, supporting the idea that bioavailability of IGFs is critical.

IGFBP5/Igfbp5a has been shown to both inhibit and potentiate IGF actions in mammalian and zebrafish cells in vitro and in vivo (*Zheng et al., 1998a*; *Zheng et al., 1998b*; *Ren et al., 2008*; *Dai et al., 2010*; *Liu et al., 2018*; *Duan and Allard, 2020*). How the same protein exerts these opposing biological effects was not well understood. The data presented in this study suggests that Papp-aa-mediated Igfbp5a proteolysis functions as a $[Ca^{2+}]$-regulated molecular switch to turn on and off IGF signaling in NaR cells. Under normal conditions, Papp-aa activity is suppressed at low levels and Igfbp5a is intact. Intact Igfbp5a acts as an IGF inhibitory protein as it binds IGF ligands and prevents its binding and activation of the IGF1 receptor. Under low $[Ca^{2+}]$ stress, Papp-a activity is increased. Papp-aa proteolytically cleaves Igfbp5a and releases IGFs from the Igfbp5a/IGF complex to activate IGF1 receptor-mediated PI3 kinase-Akt-Tor signaling, thus promoting NaR cell proliferation (*Figure 6G*). This model provides a mechanistic explanation for the seemingly opposite biological effects of Igfbp5a and IGFBP5. We suggest that when Igfbp5a/IGFBP5 is present in a microenvironment containing active Papp-aa, it potentiates IGF action because Papp-aa-mediated Igfbp5a proteolysis releases IGFs from the Igfbp5a/IGF complex. When Igfbp5/IGFBP5 is expressed in tissues without Papp-aa or with Papp-aa activity inhibited, it inhibits IGF action as Igfbp5a binds to IGFs and sequesters IGFs in the IGF/Igfbp5a complex.

While it has become evident that Igfbp5/IGFBP-5 acts as the pivot point in a switch between regulated states of inhibition and activation of IGF signaling, many questions remain. For example, mutation of Igfbp5a K148 to alanine reduced, but did not abolish Papp-aa cleavage, suggesting there may be other residues and/or other proteinases involved. Although Papp-ab can cleave Igfbp-5a efficiently in vitro, Papp-ab is clearly not involved in NaR cell reactivation in vivo as demonstrated by the *papp-ab*$^{-/-}$ mutant fish data. Whether Papp-ab-mediated Igfbp5a plays similar role in switching on/off IGF signaling in different tissues or different life stages awaits further investigation. All three pappalysin family members can proteolytically cleave Igfbp5b efficiently but the functional role of Igfbp5b and its regulations by pappalysins are unclear. We noted that Igfbp5a was partially degraded in the absence of Papp-aa. This was evident in the mock group in *Figure 3C*, *Figure 4A* and *Figure 4—figure supplement 1B*, suggesting the presence of other proteinases in the conditioned media used. This is consistent with the fact that human IGFBP5 is degraded by a host of proteinases. In addition to PAPP-A and -A2, thrombin, elastase, cathepsin G, C1s, ADAM 9, ADAM 12 s, MMP-1, and MMP-2 etc. have been shown to degrade IGFBP5 in vitro (*Duan and Allard, 2020*). Future studies are needed to elucidate whether any of these proteolytic modifications play similar roles in turning on and off IGF signaling.

Another important remaining question is how Papp-aa proteolytic activity is suppressed under normal $[Ca^{2+}]$ conditions. Since changes in external $[Ca^{2+}]$ did not change *papp-aa* mRNA level nor *igfbp5a* mRNA level, and because addition of free IGF1 or pharmacological disruption of the Igfbp/IGF complex was sufficient to activate IGF signaling and induce NaR cell reactivation under normal $[Ca^{2+}]$ conditions, we speculate that there is a $[Ca^{2+}]$-regulated post-transcriptional mechanism(s) at

work. One possible candidate is Stc. Recent studies showed that mammalian PAPP-A proteolytic activity is inhibited by STC1 and STC2 (*Jepsen et al., 2015*; *Kløverpris et al., 2015*). In this study, we showed that overexpression of STC1 or STC2 in NaR cells impaired Papp-aa-induced IGF signaling and NaR cell reactivation in vivo. Stc protein was originally discovered in bony fish as a hypocalcemic hormone (*Pang, 1973*). We now understand that the human genome has two *STC* genes while the zebrafish genome has four *stc* genes (*Yeung et al., 2012*). The mRNA levels of *stc1* but not the other *stc* genes are significantly higher in fish raised in high [$Ca^{2+}$] medium compared to those kept in low [$Ca^{2+}$] medium (*Chou et al., 2015*). Forced expression and morpholino-based knockdown of Stc1 altered NaR cell density and $Ca^{2+}$ uptake (*Tseng et al., 2009*), although these manipulations have also changed HR and NCC cell density and the uptake of other ions (*Chou et al., 2015*). Whether endogenous Stc1 and/or other Stc proteins are responsible for the [$Ca^{2+}$]-dependent suppression of Papp-aa activity awaits future studies. It is also worth noting that both human PAPP-A and zebrafish Papp-aa contain three Lin-12/Notch repeat (LNR) modules (*Boldt et al., 2004*; *Wolman et al., 2015*). LNR modules were initially discovered in the Notch receptor family (*Lovendahl et al., 2018*). These modules can bind to $Ca^{2+}$ and are involved in calcium depletion-induced Notch receptor cleavage and activation (*Rand et al., 2000*; *Vardar et al., 2003*; *Gordon et al., 2007*). The three LNR modules in human PAPP-A can bind $Ca^{2+}$ and form a $Ca^{2+}$-dependent substrate binding exosite (*Boldt et al., 2004*; *Weyer et al., 2007*; *Mikkelsen et al., 2008*). Whether these LRN modules are involved in the [$Ca^{2+}$]-dependent suppression of Papp-aa activity is unclear.

In summary, the results of this study have uncovered a novel function of Papp-aa in reactivating quiescent epithelial cells and regulating bone calcification. Papp-aa promotes quiescent NaR cells to re-enter the cell cycle by locally cleaving Igfbp5a and activating IGF signaling in a physiological context-dependent manner. These findings not only provide new insights into the physiological functions of the pappalysin family zinc metalloproteinases, but also have important implications in the regulation of cellular quiescence. Reactivation of quiescent cells is critical for tissue repair, regeneration, and adult stem cell renewal (*Cheung and Rando, 2013*; *Spencer et al., 2013*; *Yao, 2014*). Genetic studies in flies and mice suggest that insulin/IGF-PI3 kinase-Akt-Tor signaling pathway is critical in adult stem cell quiescence regulation (*Chen et al., 2009*; *Ito and Suda, 2014*; *Ziegler et al., 2015*). In *Drosophila,* neural stem cells are reactivated in response to dietary amino acids and this has been attributed to insulin produced in glia cells (*Chell and Brand, 2010*; *Sousa-Nunes et al., 2011*; *Huang and Wang, 2018*). Reactivation of the imprinted IGF2 gene expression or activation of mTOR in mouse adult hematopoietic stem cells (HSCs) promotes them to exit quiescence and proliferate (*Chen et al., 2008*; *Venkatraman et al., 2013*). IGF2 also stimulates adult neural stem cell and intestinal stem cell reactivation in mice (*Ferrón et al., 2015*; *Ziegler et al., 2015*; *Ziegler et al., 2019*). Elevated PI3 kinase-Akt-Tor activity stimulates reactivation of adult stem cells in *Drosophila* and mice (*Chell and Brand, 2010*; *Sousa-Nunes et al., 2011*; *Gil-Ranedo et al., 2019*; *Hemmati et al., 2019*). Although the role of insulin/IGF-PI3 kinase-Akt-Tor signaling in reactivating quiescent adult stem cells has been demonstrated across a wide range of species, the mechanisms underlying its cell type- and cell state-specific activation are poorly understood. The similarities among mammals, flies, and fish imply evolutionarily conserved and general mechanisms at work. Future studies are needed to determine whether the PAPP-A/A2-mediated IGFBP proteolytic cleavage plays similar roles in the reactivation of adult stem cells and/or other quiescent cells.

## Materials and methods

### Experimental animals

Zebrafish were raised and crossed according to the standard husbandry guidelines (*Westerfield, 2000*). Embryos and larvae were raised at 28.5°C in standard E3 embryo medium (*Westerfield, 2000*) unless stated otherwise. Two additional embryos media containing 0.2 mM [$Ca^{2+}$] and 0.001 mM [$Ca^{2+}$] (referred to as normal and low [$Ca^{2+}$] medium, respectively) were prepared as previously reported (*Dai et al., 2014*) and used. 0.003% (w/v) N-phenylthiourea (PTU) was added to the media to inhibit pigmentation. *Tg(igfbp5a:GFP)* fish were generated in a previous study (*Liu et al., 2017*). The *papp-aa^{p170/+}* fish were obtained from the Marc Wolman lab. All experiments were conducted in accordance with the guidelines approved by the Institutional Committee on the Use and

Care of Animals, University of Michigan and the Danish *The Animal Experiments Inspectorate* (permit numbers 2017-15-0201-01369 and 2017-15-0202-00098).

## Chemicals and reagents

Chemical and molecular biology reagents were purchased from Fisher Scientific (Pittsburgh, PA, USA) unless stated otherwise. BMS-754807 was purchased from JiHe Pharmaceutica (Beijing, China) and NBI-31772 from Tocris Bioscience (Ellisville, MO, USA). Liberase TM, Alizarin Red S, $ZnCl_2$, and batimastat were purchased from Sigma (St. Louis, MO, USA). MK2206 was purchased from Chemie-Tek (Indianapolis, IN). Rapamycin was purchased from Calbiochem (Gibbstown, NJ). Antibodies (phospho-Akt, phospho-S6, and phospho-Erk) were purchased from Cell Signaling Technology (Danvers, MA, USA). Restriction enzymes were purchased from New England BioLabs (Ipswich, MA, USA). Primers, cell culture media, antibiotics, TRIzol, M-MLV reverse transcriptase, Alexa Fluor 488 Tyramide SuperBoost Kit, and Tetramethylrhodamine (TMRM) were purchased from Life Technologies (Carlsbad, CA, USA). Anti-Digoxigenin-POD and Anti-Digoxigenin-AP antibodies were purchased from Roche (Basel, Switzerland).

## Fluorescence activated cell sorting (FACS) and qPCR

FACS was performed to isolate NaR cells from *Tg*(*igfbp5a:GFP*) fish as described (*Liu et al., 2017*). Briefly, *Tg*(*igfbp5a:GFP*) larvae were raised in E3 medium to 3 days post fertilization (dpf) then transferred to normal or low $[Ca^{2+}]$ embryo media for 18 hr. Single cell suspensions were made using Liberase TM (0.28 Wünsch units/ml in HBSS). EGFP-positive cells were sorted using FACSAria Cell Sorter (BD Biosciences, Franklin Lakes, NJ). Total RNA was isolated using TRIzol LS reagent. Total RNA was reverse transcribed using SuperScript III Reverse Transcriptase. RNaseOUT Recombinant Ribonuclease Inhibitor was added into the reverse transcript system to protect RNA from degradation. M-MLV Reverse Transcriptase was used for reverse transcription. qPCR was carried out using SYBR Green (Bio-Rad, Hercules, CA) on a StepONE PLUS real-time thermocycler (Applied Biosystems, Foster City, CA). Primers for qPCR are listed in *Supplementary file 1*.

## Generation of *papp-ab*$^{-/-}$ (Auzf1) fish lines using CRISPR/Cas9

The sgRNA targeting *papp-ab* (5'-GGGAAAGCAGCCCAGGTCGG(CGG)−3') was mixed with nCas9n-encoding mRNA in vitro transcribed from linearized pCS2-nCas9n and co-injected into wild-type embryos at the 1 cell stage as described (*Xin and Duan, 2018*). After using a subset of F0 embryos to confirm the presence of indels, the remaining F0 embryos were raised to adulthood and crossed with wild-type fish. F1 fish were then raised to the adulthood and genotyped. After confirming indels by DNA sequencing, the heterozygous F1 fish were intercrossed to generate F2 fish. The mutation is a seven nucleotide deletion (238–244 bp) in the *papp-ab* CDS, resulting in a premature stop codon in exon 1.

## Genotyping

dCAPS assay and HRMA methods were used to genotype *papp-aa* mutant. The dCAPS assay was performed following a published method (*Wolman et al., 2015*). The region containing the *papp-aa* mutation was amplified with the dCAPS primers (*Supplementary file 1*), then digested by Mse1. To genotype a large number of *papp-aa*$^{-/-}$ mutant fish, HRMA was performed as previously reported (*Liu et al., 2018*) using the *papp-aa*-HRMA primers (*Supplementary file 1*). The *papp-ab* genotyping was performed by PCR using the primers 5'-ATACGCGTCTTGACAGGCTT-3' and 5'-TAAG-CAAACCAAACCTCCGA-3' followed by Exonuclease I/FastAP Thermosensitive Alkaline Phosphatase treatment and Sanger sequencing using primer 5'-ATACGCGTCTTGACAGGCTT-3'.

## Whole-mount in situ hybridization and immunostaining

A DNA fragment encoding part of Papp-aa sequence was amplified using primers in *Supplementary file 1*. The PCR product was cloned in pGEM-T easy plasmid. The Digoxygenin-UTP labeled sense and antisense riboprobes were synthesized as previously reported (*Wang et al., 2009*). Zebrafish larvae were fixed in 4% paraformaldehyde, permeabilized in methanol, and analyzed by whole mount immunostaining or in situ hybridization analysis as described previously (*Dai et al., 2014*).

For double color in situ hybridization and immunostaining, *papp-aa* mRNA signal was detected using an anti-DIG-POD antibody (Roche), followed by Alexa 488 Tyramide Signal Amplification (Invitrogen). After in situ hybridization analysis, the stained larvae were washed in 1xPBST then incubated with a GFP antibody overnight at 4˚C. Larvae were then stained with a Cy3 conjugated Goat anti-Rabbit IgG antibody (Jackson ImmunoResearch, West Grove, PA). Images were acquired using a Nikon Eclipse E600 Fluorescence Microscope with PMCapture Pro six software.

## Plasmid and BAC constructs and BAC-mediated genetic mosaic assay

The expression plasmids for Papp-aa, Papp-ab, Papp-a2, Igfbp5a, Igfbp5a, and Igfbp3 have been previously reported (*Kjaer-Sorensen et al., 2013*; *Kjaer-Sorensen et al., 2014*). In order to generate cleavage resistant mutants of Igfbp5a and Igfbp5b, the following residues were mutated to alanine individually or in combination: Igfbp5a K147 and K148, Igfbp5b K144 and K145. Mutations were introduced by site-directed mutagenesis using QuikChange (Stratagene). The aforementioned plasmids were used as templates, and the primers used are shown in *Supplementary file 1*. All constructs were verified by sequencing analysis. The BAC constructions were generated as reported (*Liu et al., 2017*). Briefly, zebrafish Papp-aa and Papp-aa (E501A) cDNA were released from pcDNA3.1mH(+)A plasmids and sub-cloned into pIRES2-mCherry plasmid using SacII and BamH1 restriction sites. The papp-aa/papp-aa (E501A)-IRES2-mCherry-KanR cassette DNAs were amplified by PCR using primers which contain a 50 bp homology region to *igfbp5a*. The DNA cassettes were inserted into the *igfbp5a* BAC construct to replace the *igfbp5a* sequence from the start codon to the end of the first exon through homologous recombination. A similar cloning strategy was used to generate BAC(igfbp5a:hSTC1-IRES-mCherry) and BAC(igfbp5a:hSTC2-IRES-mCherry) constructs. The construction of BAC(igfbp5a: myrAkt -mCherry) was previously reported (*Liu et al., 2018*). All constructs used were confirmed by DNA sequencing at the University of Michigan DNA Sequencing Core Facility.

For Tol2 transposon-mediated genetic mosaic assay, the validated BAC DNA and Tol2 mRNA were mixed and injected into *Tg(igfbp5a:GFP)* embryos at the 1 cell stage. The embryos were raised and subjected to the low [Ca$^{2+}$] stress test as shown in *Figure 2A*. Cells co-expressing mCherry and GFP were identified and scored for cell division following a reported scoring system (*Liu et al., 2018*).

## Live imaging and microscopy

NaR cells were quantified as previously reported (*Liu et al., 2017*). To visualize NaR cells, progeny of *papp-aa$^{+/-}$;Tg(igfbp5a:GFP)* fish intercrosses were anesthetized with 0.63 mM tricaine. Live larvae were embedded in 0.3% low melting point agarose containing 0.63 mM tricaine and placed in a chamber in which the bottom was sealed with a 0.16–0.19 mm slide. The solidified agarose was then immersed in 1 ml normal [Ca$^{2+}$] medium. The bright-field and GFP fluorescent images were acquired using a Leica TCS SP8 confocal microscope equipped with HCPL APO 93X/1.30 GLYC. After images were obtained, the normal [Ca$^{2+}$] medium was washed with low [Ca$^{2+}$] medium three times. The samples were then incubated in low [Ca$^{2+}$] medium for 8 hr and imaged again. LAS X and Image J were used for image analysis.

## Morphology analysis

Body length, defined as the curvilinear distance from the head to the end of caudal tail, was measured as reported (*Liu et al., 2018*). Somite number and head-trunk angles were measured manually as reported (*Kamei et al., 2011*). Alizarin red staining was performed as previously described (*Liu et al., 2018*). Images were captured with a stereomicroscope (Leica MZ16F, Leica, Wetzlar, Germany) equipped with a QImaging QICAM camera (QImaging, Surrey, BC, Canada).

## Drug treatment

Fish were treated with BMS-754807, ZnCl2, batimastat, and other drugs as reported previously (*Liu et al., 2017*). Drugs used in this study were dissolved in DMSO and further diluted in water. ZnCl$_2$ was dissolved in distilled water. Drug treatments began at 3dpf unless stated otherwise. Drug solutions were changed daily. The samples were collected for immunostaining after 24 hr of treatment or for in situ hybridization after 48 hr of treatment. The IGF1 treatment experiment was carried

out as reported by *Alassaf et al. (2019)*. *Tg(igfbp5a:GFP)* fish were treated with recombinant fish (salmon) and human IGF-1 at 150 ng/ml (GroPrep, dissolved in 10 mM HCl and diluted in E3 embryo medium) from 3 dpf to five dpf. NaR cells were quantified as previously described (*Liu et al., 2018*).

## Cell culture

Human embryonic kidney (HEK) 293 cells were cultured in DMEM (HEK 293) supplemented with 10% FBS, penicillin, and streptomycin in a humidified-air atmosphere incubator containing 5% $CO_2$. Conditioned media were prepared as previously reported (*Duan and Clemmons, 1998*).

## Proteolytic assays and western blotting

For in vitro proteinase assay, conditioned media were harvested from HEK293 cells transfected with zebrafish Igfbp5a, Igfbp5b, Igfbp3, Papp-aa, Papp-ab, Pappa2, and Papp-aa E501A plasmid alone or together. Conditioned media from MOCK transfected cells were used as the control. The media were concentrated and used in the proteinase assay following published conditions (*Kjaer-Sorensen et al., 2014*). The proteinase assay was performed as described (*Kjaer-Sorensen et al., 2014*). Briefly, conditioned media collected from HEK293 cells transiently transfected with Papp-aa or Papp-ab was mixed in a 1:1 ratio with media containing either Igfbp5a, Igfbp5b or Igfbp3 + 100 nM IGF-II and incubated at 28°C. The reactions were stopped at different time points by adding 4x NuPAGE LDS sample buffer containing 75 mM EDTA and 4 mM PMSF. Next, proteins were separated by non-reducing SDS-PAGE and visualized by western blotting as previously described (*Boldt et al., 2001*) using monoclonal anti c-myc antibody 9E10 (Thermo Fisher) 1:1000 and HRP-conjugated rabbit anti-mouse (P0260, DAKO) 1:2000.

## Statistical analysis

Values are shown as mean ± standard error of the mean (SEM). Statistical significance between experimental groups was determined using an unpaired t test, one-way ANOVA followed by Tukey's multiple comparison test, Logrank test, or Chi-square test. Statistical significances were accepted at $p < 0.05$ or greater.

# Acknowledgements

We thank Dr. Marc Wolman, University of Wisconsin, for sharing the *papp-aa$^{p170+/-}$* fish line and Dr. Yi Xin, University of Michigan, for her help during this study. CD wishes to thank Dr. Stephen C Blacklow, Harvard University, for his helpful discussions on LNR structure and regulation. This work was supported by NSF grant IOS-1557850 and University of Michigan M-Cubed3 Project U064122 to CD, and by Independent Research Fund Denmark | Medical Sciences (DFF|FSS) Grant # 4183-00314 and Lundbeck Grant R317-2019-526) to CO. SL was supported by a fellowship from the China Oversea Scholarship Council.

# Additional information

### Funding

| Funder | Grant reference number | Author |
| --- | --- | --- |
| National Science Foundation | IOS-1557850 | Cunming Duan |
| University of Michigan | M-Cubed3 Project U064122 | Cunming Duan |
| Independent Research Fund Denmark | Medical Sciences (DFF|FSS) 4183-00314 | Claus Oxvig |
| Lundbeckfonden | R317-2019-526 | Claus Oxvig |
| China Scholarship Council | | Shuang Li |

The funders had no role in study design, data collection and interpretation, or the decision to submit the work for publication.

## Author contributions
Chengdong Liu, Shuang Li, Data curation, Formal analysis, Investigation, Visualization, Writing - review and editing; Pernille Rimmer Noer, Investigation, Visualization, Writing - review and editing; Kasper Kjaer-Sorensen, Data curation, Investigation, Writing - review and editing; Anna Karina Juhl, Investigation, Methodology; Allison Goldstein, Investigation, Visualization; Caihuan Ke, Resources, Supervision, Writing - review and editing; Claus Oxvig, Resources, Supervision, Funding acquisition, Writing - review and editing; Cunming Duan, Conceptualization, Resources, Supervision, Funding acquisition, Visualization, Writing - original draft, Project administration, Writing - review and editing

## Author ORCIDs
Chengdong Liu (iD) https://orcid.org/0000-0002-4080-8085
Cunming Duan (iD) https://orcid.org/0000-0001-6794-2762

## Ethics
Animal experimentation: All experiments were conducted in accordance with the guidelines approved by the Institutional Committee on the Use and Care of Animals, University of Michigan and the Danish The Animal Experiments Inspectorate (permit numbers 2017-15-0201-01369 and 2017-15-0202-00098).

## Decision letter and Author response
Decision letter https://doi.org/10.7554/eLife.52322.sa1
Author response https://doi.org/10.7554/eLife.52322.sa2

# Additional files

## Supplementary files
• Supplementary file 1. PCR primers used in this study.

• Transparent reporting form

## Data availability
All data generated or analysed during this study are included in the manuscript and supporting files.

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
