## [Decision Letter]

**Acceptance summary:**

This paper shows that cleavage of the IGF binding protein, Igfbp5a by Papp-aa, in zebrafish, acts to reactivate Ca^2+^ transporting epithelial cells. This in turn impact bone calcification. As such, this provides novel insight into the mechanism linking IGF signaling action on the epithelial cell to bone calcification.

**Decision letter after peer review:**

Thank you for submitting your article "The metalloproteinase Papp-aa functions as a molecular switch linking IGF signaling to adaptive epithelial growth" for consideration by *eLife*. Your article has been reviewed by three peer reviewers, and the evaluation has been overseen by a Reviewing Editor and Clifford Rosen as the Senior Editor. The following individuals involved in review of your submission have agreed to reveal their identity: Shoshana Yakar (Reviewer #1); Julian Christians (Reviewer #3).

The reviewers have discussed the reviews with one another and the Reviewing Editor has drafted this decision to help you prepare a revised submission.

General Assessment:

While it has been known for some time that plasma protein-A (PAPP-A) loss impacts bone, the mechanism has not been completely explored. The reviewers find that this paper utilizes a zebrafish model, to provide evidence that PAPP-aa (zebrafish paralog) activity in ionocytes (NaR cells, similar to human intestinal endothelial cells) is calcium dependent. They then explore an alternative mechanism by which the PAPP-A protease regulates IGF-1 bioavailability to the IGF-1 receptor.

Central conclusions:

1) They have determined that PAPP-aa protease regulates intestinal calcium transport to an extent that this alters skeletal calcification within the intact zebrafish.

2) They demonstrated that expression of PAPP-aa was required for ionocyte activation and that inhibition of PAPP-aa impaired ionocyte activation.

3) Mechanistically, these data suggest that PAPP-aa contributes to the proteolysis of the IGFBP5-IGF-1 complex, that this is co-incident with the activation of the Akt-Tor pathway and thus initiating ionocyte cell proliferation.

Essential revisions:

1) There was significant concern raised during review about specificity of PAPP-aa in the zebrafish with regards to cleavage of just Igfbp5a. The data regarding the PAPP-aa mutant clearly show that PAPP-aa is required for this activity. The prospect that PAPP-aa might be cleaving other IGFBPs and other non-Igfbp peptides was raised by multiple reviewers. A limitation of this study is that this possibility is not addressed. It was felt that alternative explanations of the results need to be more clearly presented.

2) While it was acknowledged that it is possible that PAPP-aa's mechanisms of action is via IGF-1, reviewers felt that this was not definitively demonstrated in this paper. The limitations of the data presented in Figure 6 were not fully discussed and no definitive evidence that free IGF-I is actually increased was shown. The reviewers point out that the NBI inhibitor binds to multiple IGF binding proteins as well as other proteins, including EGF binding proteins, therefore it is not a mono specific agent for IGFBP-5, particularly at the concentration used in this study. A positive control showing that IGF-I was released was considered to be important. Further, more discussion regarding the limitations needs to be provided.

3) It was felt that the data presented linking Igfbp5, IGF-1 and PAPP-aa was thin. The reviewers wished to see results from a double Igfbp5a and Papp-aa deletion mutant. The rational was that if Papp-aa acts through Igfbp5a, there should be no difference between *papp-aa^-/-^ igfbp5a^-/-^* fish and *papp-aa^+/+^ igfbp5a^-/-^* fish.

4) Expression of the constitutively active Akt mutant partially reversing the changes in the signaling pathway is reasonable evidence that the Akt pathway is involved in mediating the effects of PAPP-aa; however multiple growth factors as well as nutrients can activate the Akt pathway. IGF-I is a cofactor for multiple other growth factors and therefore simply inhibiting the IGF receptor with a relatively high concentration of a kinase inhibitor was not considered adequate specificity data. This limitation was not discussed.

5) There is one experiment wherein a cleavage site mutant of IGFBP-5a was slightly changed in terms of proteolytic degradation in the presence of activated PAPP-aa, but the data are not definitive. It was felt that an experiment showing this mutant could alter the biologic response was required.

6) There is an extensive paragraph about PAPP-A in the Discussion and an inferential comment to relating to PAPP-aa since it contains the same three modules that are important for PAPP-A activation. However, PAPP-A has different substrates therefore it's activity may not be comparable to PAPP-aa. This paragraph should be edited to deemphasize the importance of this possible correlation in light of the limitations already raised.

7) The model presented in Figure 6F is consistent with the results of the present study, but does not explain a previous study (described in the Introduction), in which deletion of Igfbp5a reduced calcified bone mass, an observation similar to deletion of PAPP-aa (Figure 2G). This previous study also described that deletion of Igfbp5a "blunted the low [Ca^2+^] stress-induced IGF signaling and NaR cell proliferation". However, at low [Ca^2+^], Igfbp5a is being proteolyzed by PAPP-aa, so why would it matter if it is deleted? Deletion of Igfbp5a would be expected to have effects opposite to those of deleting PAPP-aa (which would increase intact Igfbp5a levels). The Discussion states: "Igfbp5a is critical in activating IGF signaling and promoting adaptive epithelial growth by reactivating NaR cells" and yet the model in Figure 6F suggests that Igfbp5a maintains quiescence. There should be some acknowledgement/explanation of this contradiction.

8) There is mention of the rescue of the PAPP-aa mutants with wild-type PAPP-aa, but no mention of whether this rescued the lethal phenotype.

9) The impact statement should be reworded, as it is not clear what "quiescence exit decision" means without reading the paper.

---

## [Author Response]

Essential revisions:1) There was significant concern raised during review about specificity of PAPP-aa in the zebrafish with regards to cleavage of just Igfbp5a. The data regarding the PAPP-aa mutant clearly show that PAPP-aa is required for this activity. The prospect that PAPP-aa might be cleaving other IGFBPs and other non-Igfbp peptides was raised by multiple reviewers. A limitation of this study is that this possibility is not addressed. It was felt that alternative explanations of the results need to be more clearly presented.

We agree that demonstrating the specificity is critical. To address this concern, we have performed additional proteinase assays. As shown in Figure 4A, Papp-aa efficiently cleaved IGFBP5 and IGFBP4, whereas it did not cleave IGFBP1, -2, -3, and -6. This finding is in good agreement with published mammalian studies (Oxvig, 2015; Conover and Oxvig, 2018). Zebrafish genome does not contain any *igfbp4* gene, but it has 2 igfbp5 genes and several other igfbp genes. We mapped the expression profiles of these *igfbp* genes in NaR cells. The new results, shown in Figure 4D, showed that *igfbp2a* and *igfbp5a*mRNA is highly expressed in NaR cells. The levels of other *igfbp* genes are much lower. Among the 3 pappalysin family members, Papp-aa and Papp-ab can both proteolytically cleave Igfbp5a, while Papp-a2 had little activity (Figure 3C). In comparison, Igfbp5b can be cleaved by all 3 pappalysin family members (Figure 4B) and Igfbp3 can be degraded by Papp-a2 only, albeit at very low levels (Figure 4C). These data together with the data shown in Figure 1, suggest that Igfbp5a is likely an important substrate of Papp-aa in NaR cells.

To further address the specificity issue in vivo, we generated a new mutant fish line by deleting the paralogous *papp-ab* gene using CRISPR/Cas9. The new results shown in Figure 2—figure supplement 2, showed that genetic deletion of Papp-ab had no effect on low [Ca^2+^] stress-induced NaR cell reactivation and proliferation. These new data argues strongly that Papp-aa-mediated cleavage of Igfbp-5a is critical in NaR cells.

2) While it was acknowledged that it is possible that PAPP-aa's mechanisms of action is via IGF-1, reviewers felt that this was not definitively demonstrated in this paper. The limitations of the data presented in Figure 6 were not fully discussed and no definitive evidence that free IGF-I is actually increased was shown. The reviewers point out that the NBI inhibitor binds to multiple IGF binding proteins as well as other proteins, including EGF binding proteins, therefore it is not a mono specific agent for IGFBP-5, particularly at the concentration used in this study. A positive control showing that IGF-I was released was considered to be important. Further, more discussion regarding the limitations needs to be provided.

We appreciate this concern. As shown in Figure 6, addition of NBI-31772 was sufficient to increase Akt and Tor activity and promote NaR cell reactivation under normal [Ca^2+^] conditions. This effect of NBI-31772 is clearly mediated by IGF signaling because 1) NBI-31772 increased Akt and Tor signaling activity in an IGF1 receptor-dependent manner (Figure 6C and 6D), and 2) NBI-31772 treatment-induced NaR cell proliferation was abolished by co-treatment with inhibitors blocking the IGF1 receptor, PI3 kinase, Akt, and Tor activity (Figure 6E; Figure 6—figure supplement 1). Furthermore, NBI-31772 treatment did not alter pErk signaling and MEK inhibitor U0216 did not inhibit NBI-31772-induced NaR cell proliferation (Figure 6—figure supplement 1). These data suggest that the effect of NBI on NaR cells is mediated by IGF1 receptor-PI3 kinase-Akt-Tor signaling.

We agree that the NBI inhibitor is not a mono specific agent for IGFBP-5. Our new gene expression analysis results (Figure 4D) show that in addition to Igfbp5a, Igfbp2a is also highly expressed in NaR cells. The levels of *igfbp1b, igfbp3, igfbp5b,* and *igfbp6b*mRNA in NaR cells are much lower, while those of *igfbp2b* and *igfbp6a* are barely detectable. In the revised manuscript, we have discussed the possibility that NBI may also affect other Igfbps (Discussion paragraph four).

Zebrafish has 4 IGF ligand genes (Zou et al., 2009). Currently, antibodies against these IGFs are not available and there are currently no tools to measure free IGFs in these or other cells in vivo. We have discussed these limitations in the revised manuscript (Discussion paragraph four). To circumvent this problem, we took an alternative approach following a recent paper by Alassaf et al., 2019. We found that adding free human or fish IGF1 was sufficient to activate NaR cell proliferation (Figure 6F and Figure 6—figure supplement 2). These new data have provided additional evidence supporting the idea that bioavailable IGFs are important.

3) It was felt that the data presented linking Igfbp5, IGF-1 and PAPP-aa was thin. The reviewers wished to see results from a double Igfbp5a and Papp-aa deletion mutant. The rational was that if Papp-aa acts through Igfbp5a, there should be no difference between papp-aa^-/-^ igfbp5a^-/-^ fish and papp-aa^+/+^ igfbp5a^-/-^ fish.

Following this suggestion, we have obtained *papp-aa^-/-^* and *igfbp5a^-/-^* double mutant embryos and compared them to *papp-aa^-/-^* single mutant embryos. As shown in Figure 4E, no differences in NaR cell reactivation were found between *papp-aa^-/-^*single and *papp-aa^-/-^: igfbp5a^-/-^* double mutant fish.

We also measured pAkt levels in these fish and found no difference. These data results are presented in Figure 5—figure supplement 1B.

4) Expression of the constitutively active Akt mutant partially reversing the changes in the signaling pathway is reasonable evidence that the Akt pathway is involved in mediating the effects of PAPP-aa; however multiple growth factors as well as nutrients can activate the Akt pathway. IGF-I is a cofactor for multiple other growth factors and therefore simply inhibiting the IGF receptor with a relatively high concentration of a kinase inhibitor was not considered adequate specificity data. This limitation was not discussed.

We agree. We have also discussed the limitation of the CaAkt results (Discussion paragraph three). As mentioned above, to further test our idea, we treated zebrafish embryos with free fish and human IGF1. The new data show that adding free IGF1 was sufficient to stimulate NaR cell proliferation under normal calcium conditions (Figure 6F and Figure 6—figure supplement 2). This has provided new and independent evidence supporting our conclusion.

5) There is one experiment wherein a cleavage site mutant of IGFBP-5a was slightly changed in terms of proteolytic degradation in the presence of activated PAPP-aa, but the data are not definitive. It was felt that an experiment showing this mutant could alter the biologic response was required.

We appreciate this comment. We have repeated the proteinase assays and the new results showed that mutation of Igfbp5a K148 to Alanine indeed reduced Papp-aa-mediated proteolysis (Figure 4—figure supplement 1). But the Igfbp5aK148A mutant was also partially degraded, suggesting there may be other proteinases and additional cleavage site(s). This is not unexpected. In addition to PAPP-A and A2, other proteases have been reported to degrade IGFBP-5 in vitro, including thrombin, elastase, cathepsin G, C1s, ADAM 9, ADAM 12s, MMP-1, and MMP-2 etc. (Clemmons, Endocr Rev. 2001, 22:800-17; and Duan and Allard, 2020). Although human PAPP-A-mediated human IGFBP-5 cleavage has been reported and the cleavage site(s) mapped for two decades, to date there is no biological data on any PAPP-A-resistant IGFBP-5 mutants. This is in large part due to the fact that many proteinases can cleave IGFBP-5, depending on different locations/stages, cell types, and physiological/pathological states. In this study, we found that recombinantly expressed Igfbp5a is unstable and susceptible to degradation. Even in the absence of Papp-aa or Papp-ab, it degrades quite rapidly. This is evident in the Mock group shown in Figure 3C, Figure 4A, and Figure 4—figure supplement 1B. Because of this, we were unable to obtain sufficient amount Igfbp5a for biological assays. In the revised manuscript, we have discussed these limitations and the need for further experiments in the future (Discussion paragraphs four, five and six). We have also moved the Igfbp5a and Igfbp5b mutant data into the supplemental materials.

6) There is an extensive paragraph about PAPP-A in the Discussion and an inferential comment to relating to PAPP-aa since it contains the same three modules that are important for PAPP-A activation. However, PAPP-A has different substrates therefore it's activity may not be comparable to PAPP-aa. This paragraph should be edited to deemphasize the importance of this possible correlation in light of the limitations already raised.

We agree that there is no experimental evidence showing the three structurally conserved LNR modules in zebrafish Papp-aa play a similar role as those in human PAPP-A. In the revised manuscript, we have shortened and reworded the Discussion about LNR modules (paragraph seven).

7) The model presented in Figure 6F is consistent with the results of the present study, but does not explain a previous study (described in the Introduction), in which deletion of Igfbp5a reduced calcified bone mass, an observation similar to deletion of PAPP-aa (Figure 2G). This previous study also described that deletion of Igfbp5a "blunted the low [Ca^2+^] stress-induced IGF signaling and NaR cell proliferation". However, at low [Ca^2+^], Igfbp5a is being proteolyzed by PAPP-aa, so why would it matter if it is deleted? Deletion of Igfbp5a would be expected to have effects opposite to those of deleting PAPP-aa (which would increase intact Igfbp5a levels). The Discussion states: "Igfbp5a is critical in activating IGF signaling and promoting adaptive epithelial growth by reactivating NaR cells" and yet the model in Figure 6F suggests that Igfbp5a maintains quiescence. There should be some acknowledgement/explanation of this contradiction.

Our data and model (Figure 6G) suggest that Papp-aa cleaves Igfbp5a in a [Ca^2+^]-dependent manner. Under normal [Ca^2+^] conditions, Papp-aa activity is suppressed and Igfbp-5a is intact. Intact Igfbp5a acts as an IGF inhibitory protein as it binds IGF ligands and sequester the IGF ligands from the IGF1 receptor. Under low [Ca^2+^] stress, Papp-a activity is increased. This increases IGFBP-5a proteolytic cleavage and releases IGFs from the IGFBP-5a/IGF complex to activate IGF-1 receptor and promotes NaR cells to proliferate. We propose that IGFBP-5 is part of a molecular switch that turns IGF signaling on or off in NaR cells. We have revised the text, Figure 6G and the legends to clearly explain these findings and our model to avoid any confusions.

The results of this study and those of Liu et al., 2018, are in good agreement with each other. In both *papp-aa^-/-^* mutant and *igfbp5a^-/-^* mutant fish, the low [Ca^2+^] stress-induced IGF signaling and NaR cell proliferation were blunted.These mutant fish also both suffered from Ca^2+^ and die prematurely, but *papp-aa^-/-^*mutant appeared to be more severe and die younger, indicating that Papp-aa may have substrates other than Igfbp5a. In this study, we have presented data showing that Papp-aa also cleaved Igfbp5b (Figure 4B and Figure 4—figure supplement 1C). We have also added new data comparing *papp-aa^-/-^*mutants and the *papp-aa^-/-^* and *igfbp5a^-/-^* double mutants (see above).

8) There is mention of the rescue of the PAPP-aa mutants with wild-type PAPP-aa, but no mention of whether this rescued the lethal phenotype.

The Tol2 transposon-mediated genetic mosaic assay faithfully target the expression of Papp-aa in a subset of NaR cells in a random fashion. While this approach is very useful in testing the sufficiency of Papp-aa and its mode of action, it cannot rescue the lethality because it only targets the expression of Papp-aa transiently and to a small subset of NaR cells in a mosaic pattern. We have added a sentence in the revised manuscript to make this point clearer (subsection “Endogenous Papp-aa proteinase activity in NaR cells is critical”).

9) The impact statement should be reworded, as it is not clear what "quiescence exit decision" means without reading the paper.

Following this suggestion, we have re-worded the impact statement.